# Hybrid-Lambda: A low specific rating rotor concept for offshore wind turbines

Daniel Ribnitzky[1], Frederik Berger[1], Vlaho Petrović[1], and Martin Kühn[1]

[1]ForWind - Center for Wind Energy Research, University of Oldenburg, Institute of Physics, Küpkersweg 70, 26127 Oldenburg, Germany

**Correspondence:** Daniel Ribnitzky (daniel.ribnitzky@uol.de)

**Abstract.** We introduce an aerodynamic rotor concept for an offshore wind turbine which is tailored for an increased power feed-in at low wind speeds by a substantial increase of the rotor diameter while maintaining the rated power. The main objective of the conceptual design is to limit the steady-inflow loads (blade flapwise root bending moment (RBM) and thrust) to the maximum values of a reference turbine. The outer part of the blade (i.e. outer 30% span) is designed for a higher design tip speed ratio (TSR) and a lower axial induction than the inner part. By operating at the high TSR in light winds, the slender outer part fully contributes to the increased power capture. In stronger winds the TSR is reduced and the torque generation is shifted to the inner section of the rotor. Moreover, the blade design efficiently reduces the power losses when the flapwise RBM is limited through peak shaving, below rated wind speed. This is of high importance, given the wind speed distribution at offshore sites. The characteristics of the rotor are first investigated with stationary blade element momentum simulations and further analyzed with aeroelastic simulations, considering the flexibility of blades and tower to show that a structural design is feasible even for a blade of this size and complexity. The economic revenue and the cost of valued energy of the turbine is estimated and compared to the IEA 15 MW offshore reference turbine, considering a fictitious wind speed-dependent feed-in price. Our results for the turbine concept with an increase of rotor diameter by 36% show that the revenue can be increased by 30% and the cost of valued energy can be reduced by 16% compared to the reference turbine.

## 1 Introduction

In the last decades, wind turbines have been designed extremely successfully to reduce the levelized cost of energy (LCoE) of wind power. With the rise of the proportion of wind power in the energy system, the market value of wind power decreases during periods of strong winds and the exchange price for wind power can be close to zero on windy days, as expected by May et al. (2015). López Prol et al. (2020) named this effect as self-cannibalisation of wind power and emphasised that there is a need for a change of mindset in wind turbine design. Future wind turbine design should focus more on improving the value of wind power in the entire energy supply system. This should favour more steady power feed-in, higher capacity factors, reduced power forecast errors and a better utilization of the transmission system.

In the past few years, a clear trend can be observed in reducing the specific rating (i.e. the ratio of rated power to rotor swept area) for newly installed turbines, especially onshore as reported by Hand et al. (2018) and Bolinger et al. (2021). Typically, such modern three-bladed onshore turbines operate with a high design tip speed ratio (TSR) of up to 10 and a high power coefficient in the lower partial load range, extending from cut-in wind speed to approximately $7.5$ to $9 \mathrm{~m~s}^{-1}$. The high aerodynamic efficiency matches perfectly with the dominant range of the wind speed distribution at light to medium wind sites. Due to the high TSR, such turbines reach the rated rotor speed at the aforementioned wind speed, which is well below the rated wind speed. Consequently, the power coefficient $c_p$ and thrust coefficient $c_T$ decrease significantly before rated power is reached (at approx. $10$ to $11 \mathrm{~m~s}^{-1}$) which results in considerable power losses, as described in Gasch and Twele (2012). However, the high design TSR results in slender blades and the reduced $c_T$ values facilitate load-reduced and cost-efficient blade designs. Consequently, the swept rotor area can be significantly increased compared to traditional designs. All in all, such an onshore design philosophy resulted in significant gains in annual energy production (AEP) and higher capacity factors, which dramatically lowered the LCoE in recent years.

Several investigations (e.g. Hirth and Müller (2016), Johnson et al. (2021), Wiser et al. (2021)) state that turbines with larger swept rotor areas in relation to rated power are beneficial for the energy system. Thus, an increased and more steady power feed-in at low wind speeds would be desirable. In contrast, the power feed-in at strong wind speeds could be partially reduced, as sufficient power will be available from the increasing number of conventional wind energy converters. Recent studies address wind turbine concepts with very low specific ratings for onshore sites. The *LowWind Turbine*, developed at DTU by Madsen et al. (2020), with a specific rating of $100 \mathrm{~W~m}^{-2}$ has a uniquely low cut-out wind speed of $13 \mathrm{~m~s}^{-1}$. The physical challenge of resisting increased loads is at least partially avoided, as the concept does not operate at high wind speeds. Swisher et al. (2022) discuss the economic competitiveness of such an onshore turbine. Further innovative concepts for very large rotors are discussed by Johnson et al. (2019) within the *big adaptive rotor* project. Increasing the power output in light winds but decreasing the loads in strong winds contradicts basic physical principles. A large swept rotor area would be needed in light winds, while the rotor area should be reduced to limit the loads in stronger winds. Concepts that mechanically adjust the rotor area have a long history but never reached the level of technical feasibility on a large scale. Jamieson et al. (2005) approached this problem with telescopic blades. Among others, Agarwala and Ro (2015) introduced a separated pitch system for the blade tip and Feil et al. (2020) investigated trailing edge flaps. Noyes et al. (2020) and Qin et al. (2020) designed and discussed downwind morphing rotors. All these ideas have in common that additional actuators increase the mass and costs of the blade and introduce a complex dynamic blade response. Consequently, there is a need to approach this design problem with pure aerodynamic and control design tools without any additional mechanical actuators. Lowering the axial induction and sacrificing a high power coefficient to favour larger rotor diameters is a frequently discussed approach. Jamieson (2020) took the idea of the low induction rotor developed by Chaviaropoulos and Sieros (2014) one step further and derived an optimal axial induction distribution over the blade span, which allows for greater power gain with a modest increase in rotor diameter, compared to a constant low induction factor of one fifth. Designing a blade with variable design conditions along the blade span is sometimes proposed. Wobben (2001) issued a patent for a blade with a step in the radial distribution of the

design TSR. This concept follows the objective of reducing unintended stall effects on the blade of a variable-speed turbine in gusty winds. It was not used to enable large rotors with low specific ratings, as pointed out with the *Hybrid-Lambda* concept.

We want to take the idea of low specific power wind turbines one step further to offshore applications. But the transfer of the current onshore design approach to offshore sites would be inherently uneconomic. With the usage of large blades comes the need for peak shaving (e.g. pitching to feather below rated power) to limit the flapwise root bending moment (RBM). Many blade designs are optimized for one single operational point, usually zero pitch at design TSR. Thus, peak shaving comes with great losses in the power coefficient. These losses significantly reduce the AEP as they occur close to rated wind speed, where most offshore sites see a very high wind speed probability in the Weibull distribution. The higher annual average wind speed and the increased design loads at offshore sites traditionally favour higher specific ratings if a low LCoE is the design driver. For the period of 1995 to 2020, Borrmann et al. (2018) report a modest decrease in specific rating from $450 \mathrm{~W} \mathrm{~m}^{-2}$ to $350 \mathrm{~W} \mathrm{~m}^{-2}$ for European offshore wind farms, but the study also shows a significant variance in the specific rating. It is unclear whether future offshore projects will request a further reduction of specific ratings in favour of the energy system or prefer medium specific ratings and increased rated power to optimize LCoE on the wind farm level.

We postulate that more system-friendly offshore turbines with dramatically lower specific ratings are required for the large-scale exploitation of offshore wind energy and the decarbonisation of the electricity sector in the 2030ies and 2040ies. Therefore, we introduce an innovative concept with an increased power feed-in at light winds, which still allows for further turbine operation at higher wind speeds by finding a good compromise between load reduction and aerodynamic efficiency.

The objective of this paper is the introduction of an innovative aerodynamic rotor concept for offshore wind turbines, which allows, in combination with advanced turbine control strategies, the above-mentioned modifications in the power curve characteristics. As a reference turbine, we use the IEA 15 MW offshore wind turbine, designed by Gaertner et al. (2020). The main objectives of the conceptual design are twofold. Firstly, the steady-inflow loads (blade flapwise RBM and thrust) are limited to the maximum values of the reference turbine while greatly increasing the swept rotor area to capture more energy in light wind conditions. Secondly, instead of aerodynamically optimizing the blades for the operation below rated wind speed and below the maximum allowable loads, a blade design should already take the application of peak shaving into consideration. The design process should be seen as a compromise finding between power maximization below the limiting loads and minimization of losses when peak shaving is applied. We want to introduce a design methodology for blades that are aerodynamically optimized for peak shaving. The aim is to address challenges in the design process concerning aerodynamics, structural design and control strategies. We want to pinpoint the advantages of the conceptual design in transient load cases as well as design driving dynamic loads and we address the economic competitiveness of the resulting turbine concept.

This paper is based on the design methodology and steady-state aero-static investigations of the general concept introduced by Ribnitzky et al. (2022). The current publication includes a more comprehensive development of the concept, an aero-servo-elastic optimization for the blade and tower design, a dynamic analysis of the turbine including controller design, a reduced set of design load cases, an investigation of transient extreme wind shear events and a techno-economic evaluation.

## 2 Methodology

This section first describes the aerodynamic concept. The second subsection explains the methodology that is used to design the rotor and to simulate the aerodynamics and loads.

### 2.1 Aerodynamic concept: The Hybrid-Lambda Rotor

The main idea of the concept is to dramatically increase the rotor swept area. This is accompanied by designing the outer part of the blade (e.g. outer 30% of the rotor) for a higher TSR (compared to the inner part) and reducing the design axial induction factor in the outer region, resulting in a much slender outer section. The design methodology can be applied to any given wind turbine rotor by adjusting the main design variables, namely the specific rating, the TSRs for the inner and outer part of the blade, the spanwise position of the transition between the two design regions, and the desired axial induction factors for the two blade regions (see Sect. 3.1). In this study, we applied the methodology to a 15 MW offshore wind turbine to simplify the understanding of the concept and to discuss a use case. The IEA 15 MW reference turbine serves as a basis for the design.

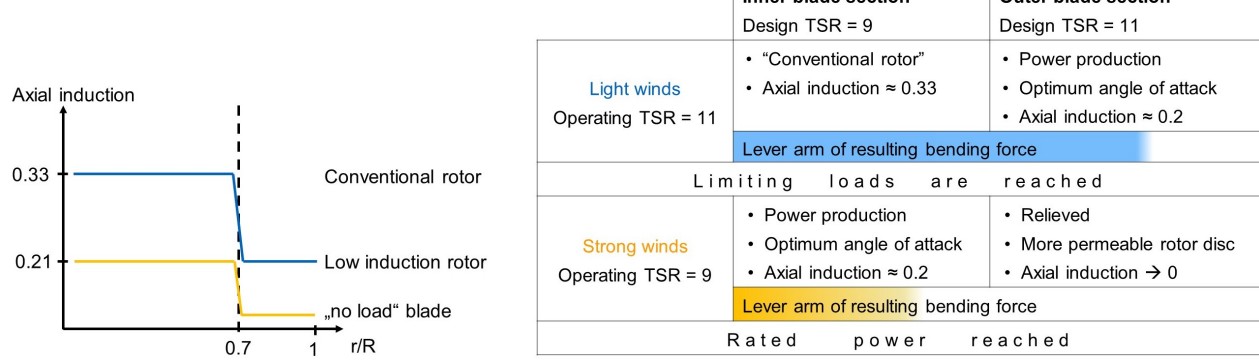

**Figure 1.** Left: Desired axial induction along the blade span for the operating modes in light wind (blue) and strong wind (yellow). Right: Schematic overview of the design criteria of the *Hybrid-Lambda* concept.

The rotor is designed to operate in a light wind and a strong wind mode. Figure 1 visualizes the desired axial induction in the two operating modes and provides an overview of the characteristic key points for the two parts of the rotor.

In light wind conditions, the rotor operates at the high TSR of 11 and the slender outer part fully contributes to the increased power capture. The outer part is now operating in its design point, defined as the high TSR, an axial induction factor of 0.21 and the optimal angle of attack (the latter is here derived from the optimal lift-to-drag ratio). The outer part can thus be interpreted as a low induction rotor extension. The inner part of the rotor operates like a conventional rotor with an axial induction factor close to 0.33. This is chosen in order to maximize the power output in light winds. But the reader should bear in mind that this part is not operating at its design point, as it is designed for a lower TSR of 9.

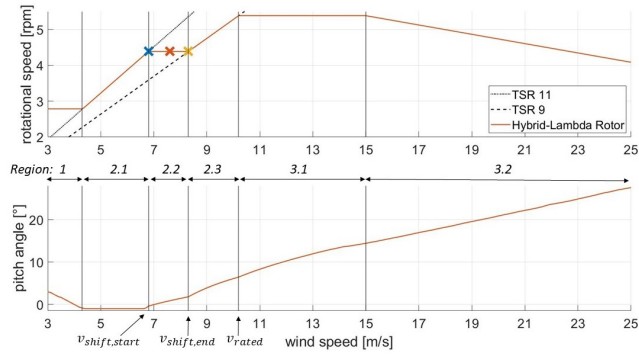

**Figure 2.** Rotational speed and pitch schedule over wind speed, crosses indicate operational points shown in Fig. 8 with corresponding colours

In stronger winds (but still below rated wind speed), the design value of the stationary RBM is reached. Then the TSR is reduced to a value of 9 and the torque generation is shifted to the inner section of the rotor, which is now operating in its design point, defined as the low TSR, an axial induction factor of 0.21 and the optimal angle of attack (AoA). In contrast, the outer region is significantly relieved, as it is now operating at very low angles of attack. Ideally, the outer part of the rotor would now operate at an axial induction of zero which means it is no longer contributing to the power production and would not produce any thrust loads. Of course, this is not feasible, but it should serve as an objective. In this way, the outer part of the rotor disc gets more permeable and the lever arm of the resulting bending force is reduced. As the rotor is designed for and operated at two different TSR, we refer to the concept in the following as the *Hybrid-Lambda Rotor*.

The transition between the operating modes introduces a new control region since the switching of the TSR is not a sudden change rather than a continuous reduction in TSR. In this paper, it is realized with a constant rotational speed (rpm) in region 2.2 as shown in Fig. 2. The reduction in TSR alone (with a constant rpm) is not enough to limit the loads. On the contrary, it is part of the design methodology to combine a reduction in TSR and pitching to feather for load limitation as further analysed in Sect. 3.1. Consequently, the so-called strong wind mode can not be described with a constant pitch angle. With increasing wind speed the pitch angle is gradually increased towards feather to limit the flapwise RBM. This action will be referred to as peak shaving in the following. Note, that the transition of TSRs could also be realized in different ways (e.g. sudden reduction or gentle increase in rpm). In fact, the optimal combination of TSR and pitch for the transition region can be found by constraining the flapwise RBM and searching for the optimum in the matrix of power coefficients for all relevant TSR and pitch values. These optimization routines resulted in a gently increasing rpm throughout region 2.2 and 2.3. However, for all wind speed bins, the increase in the power output was never larger than 0.5% of rated power compared to the constant rpm solution presented here. Consequently, the aforementioned alternative for the transition region is not presented in this paper.

Furthermore, we design the blade in a way that peak shaving is applied more efficiently. The inner section is designed with a twist offset towards stall. This comes with several advantages. The inner section does not operate in the design point in the low wind regime. As it is twisted towards stall and operated at a higher TSR than it was designed for, a fairly conventional induction factor of 0.33 can be reached, which leads to an increase in the power coefficient in the low wind regime. The angle for the twist offset is derived iteratively in stationary blade element momentum (BEM) simulations to reach the desired axial induction factor of 0.33 in the inner section at the high TSR. Using the twist instead of the chord length as a tool for this increase in the axial induction factor allows to use smaller chord lengths which leads to more slender, lighter and possibly cheaper blades. Hence, the twist offset defines the difference of the axial induction factor between the light and strong wind mode for the inner part of the blade and it further influences the pitch angle at $v_{shift,end}$ that is needed to limit the loads. In fact, the pitch angle of 2.2° at $v_{shift,end}$ almost perfectly counterbalances the twist offset of -2.5°. Hence, the inner part of the blade operates in it's optimal lift to drag ratio at this wind speed, although the entire blade is already pitched to feather for load reduction. When peak shaving is applied, pitching shifts the inner section to operate at its aerodynamic optimum rather than moving away from it. It reaches its design point (an induction factor of 0.21 at the low TSR), which is beneficial for load reduction. In contrast, the outer section is now operated in a "pitched-to-feather-condition" and is greatly relieved. The limits to this methodology are negative lift and the stall angle. The latter is also plotted in Fig. 8.

The overall design and optimization workflow is illustrated in Fig. 3. The process can be explained in four steps: An aerodynamic blade optimization, an aero-structural optimization of the blade, a structural optimization of the tower and the aero-servo-elastic simulations. In the first step (aerodynamic optimization), the design variables are the transition point between the inner and outer blade section, the design TSRs, the design axial induction factors, the twist offset and the design angle of attacks. Once a reasonable design is established the influence of the rotor radius is investigated. In the second step (the aero-structural optimization), the design variables are the airfoil positions and the spar cap thickness. When this step is converged the aerodynamic optimization is re-calculated once with the new airfoil positions. As a third step, the tower and monopile are optimized for a fixed rotor design. The resulting turbine design is then investigated in aero-servo-elastic simulations.

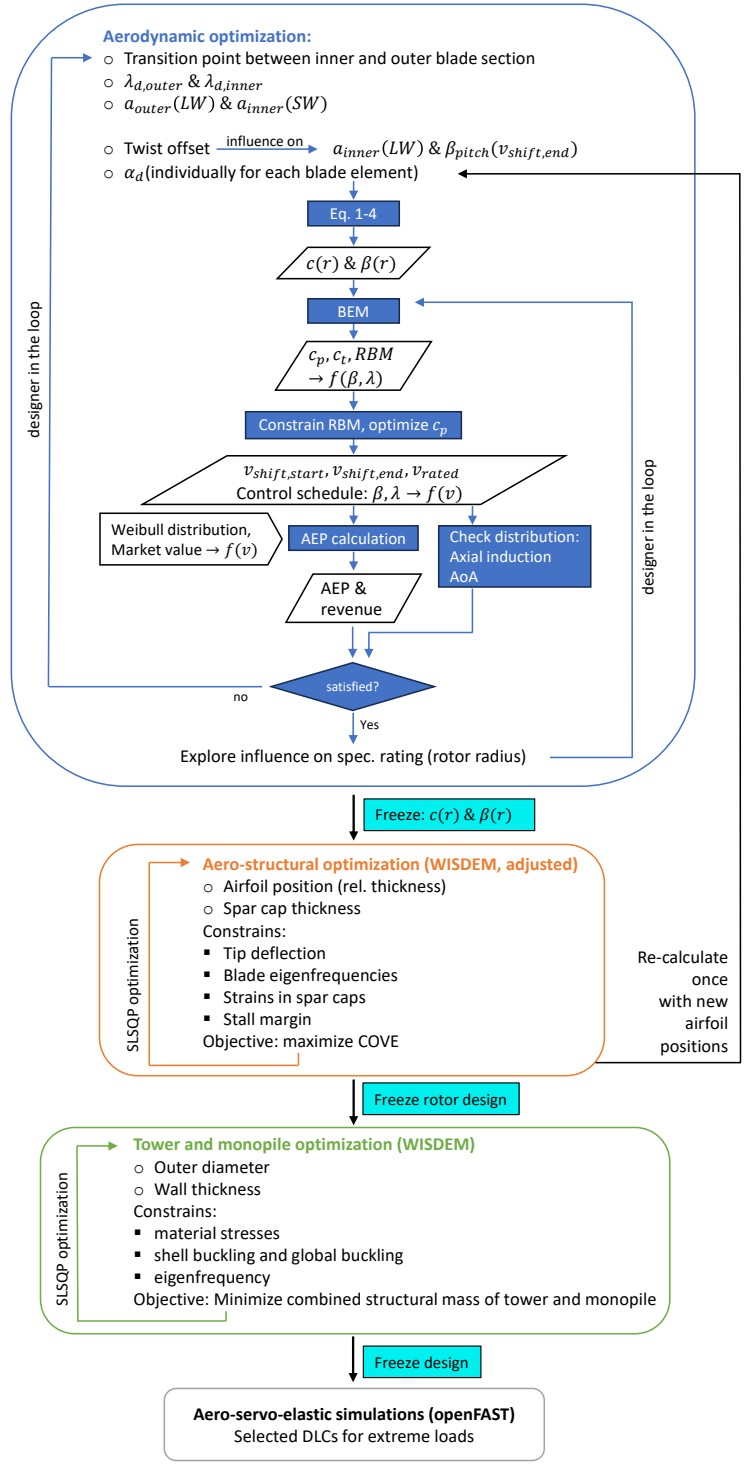

**Figure 3.** Design and optimization work flow of the *Hybrid-Lambda* concept, round bullet points: Free design variables, squared bullet points: Constraints, diamonds: Outputs, $f(...)$ : As a function of (...), LW: Light wind, SW: Strong wind

## 2.2 Design and simulation methodology

In the first step in the blade design, we calculated the chord and twist distribution along the blade span. As the *Hybrid-Lambda Rotor* is compared with the IEA 15 MW reference turbine, the same airfoil family is used and the airfoil distribution along blade span is adopted in a first step. The airfoil position is later optimized as described in Sect. 2.3. With the target design TSR ($\lambda_d$), the design axial induction factor ($a$) and the design AoA ($\alpha_d$), the corresponding chord ($c$) and twist distribution ($\beta$) for the two blade sections are calculated, following a procedure described by Burton et al. (2011). We chose the design AoA and the corresponding lift coefficient ($C_l$) as the angle with the highest lift-to-drag ratio individually for each blade station. This results in a slightly discontinuous chord and twist distribution. Therefore, smoothing with a moving average window is applied. First, the tangential induction factor ($a'$) from Eq. (1) is calculated. The distribution of the inflow angle ($\phi$) can be calculated as written in Eq. (2) and the twist distribution results according to Eq. (3). Finally, the chord distribution ($c$) is calculated with Eq. (4).

$$a' = \frac{a(1-a)}{\lambda_d^2(\frac{r}{R})^2} \tag{1}$$

$$\tan\phi = \frac{1-a}{\lambda_d(\frac{r}{R})(1+a')} \tag{2}$$

$$\beta = \phi - \alpha_d \tag{3}$$

$$c = \frac{8\pi R\lambda_d(\frac{r}{R})^2 a'}{BC_l\sqrt{(1-a)^2 + \left(\lambda_d(\frac{r}{R})(1+a')\right)^2}} \tag{4}$$

Here, $R$ is the radius of the turbine, $r$ is the mean radius of the local blade element and $B$ is the number of blades.

In the second step, the twist offset towards stall is applied to the inner section and smoothing of the chord and twist distribution is applied where necessary. In the root section, the non-dimensional chord $c/R$ is adopted from the reference turbine (which will be necessary to carry the increased torsional and edgewise bending moments) and the maximum twist is limited to $15°$. The design process continues with iterations on the axial induction distribution and the twist offset to meet the requirements regarding the angle of attack and axial induction distribution in both operating modes.

To iterate on the design process, we investigate the derived design concepts as rigid structures with steady-state BEM simulations (as described by Hansen (2008) including Prandtl tip-loss and root-loss corrections as well as the Glauert high thrust correction with the approximation by Buhl (2005). The full set of relevant TSRs and pitch angles is computed and in a second step, the control schedule is defined, e.g. assigning the specific TSR and pitch angle over the operational range of wind speeds.

We iteratively optimize the blade design on the basis of the BEM simulations. After the design criteria are satisfied, we continue investigating the rotor concept with aeroelastic simulations, as described in the next section.

## 2.3 Structural design, aeroelastic simulation and optimization methodology

To further investigate the feasibility of the *Hybrid-Lambda Rotor* we develop a structural model for the blade. The workflow
described in this section is carried out after freezing the design output-variables rotor radius and the chord and twist distribution. A link back to the aerodynamic optimization was only performed for a few major design versions, as indicated in Fig. 3. An initial layup, which is close to a scaled layup of the IEA 15 MW reference turbine, is designed in *NUMAD* (Berg and Resor, 2012), consisting of a blade shell, two spar caps, two shear webs, leading edge and trailing edge reinforcements. Blade cross-section properties are calculated with *PreComp* (Bir, 2005). In the next step, the blade is optimized with the *Wind-plant Integrated*
*System Design and Engineering Model - WISDEM* (Dykes et al. (2021)). This open-source tool allows for rapid design space exploration, including the calculation of characteristic curves, steady-state load calculations, material stress assessments and optimization routines. The source code of WISDEM is adjusted to implement the control strategies of the *Hybrid-Lambda Rotor*, as described in the following. For each design iteration, the wind speed $v_{shift,start}$ at which the transition from the light wind to the strong wind mode should start is calculated first. This is done by finding the operational point at maximum
power coefficient for the given turbine design (at $TSR = 11$ and $fine\ pitch = -0.8°$) when the limiting flapwise RBM is first reached. For higher wind speeds, the rotational speed schedule is adjusted in order to execute the transition to the low TSR. Further, a peak shaving algorithm ensures the limitation of the flapwise RBM. In a steady-state load calculation, the strains in the spar caps are calculated according to Hansen (2008) and the maximum tip deflection is derived. Free design variables are the radial positions of the airfoils for the inner blade section (airfoils for the outer blade section are locked) and the spar cap
thickness on the suction and pressure side. Constraints for the optimization process are tip deflection, blade eigenfrequencies (must be above the rated blade passing frequency, 3P), the strains in the spar caps and a stall margin. The latter would only be active if the change in the airfoil position leads to an operating angle of attack larger than the stall angle of the respective airfoil (chord and twist are not optimized in this structural design step). The objective function of the optimization process is the cost of valued energy. For each iteration the schedule of rpm, pitch, power, thrust and flapwise RBM over wind speed
is re-calculated. The considered load case for the constraints is a steady inflow at the strongest wind speed in the light wind mode $v_{shift,start}$, as calculated for each design iteration (in this case $v = 6.9\ \mathrm{m\ s^{-1}}$, TSR $= 11$, $\beta_{\mathrm{pitch}} = -0.8°$). This is the operational point just before the limitation of the flapwise RBM starts by lowering the TSR and pitching, and it is found to be the most critical regarding steady-inflow tip deflection and RBM (compared to other steady inflow wind speeds). To account for higher loads that will certainly occur under dynamic inflow load cases, the constraints for tip deflection and strains are
set relatively strict. This way, the optimization routine can run computationally efficiently and more complex load cases are verified afterwards.

The tower of the turbine is optimized with *WISDEM*, too. As the main focus of this paper is the aerodynamic rotor concept, we only present a preliminary tower design and the simple choice of a monopile foundation was made, although the authors are well aware that in reality, such a large turbine will most likely be mounted on a jacket substructure. Design variables for the optimization are the tower and monopile diameter and the material thickness. Constraints are the material stresses, shell buckling and global buckling and the first structural natural frequency should be between 1P and 3P excitation (with 10% safety margin) within the operational range of the rotor speed. The objective function of the optimization routine is the combined structural mass of tower and monopile. For the optimization, a constant loading is applied at the tower top. This loading is derived with an initial aeroelastic simulation with an extreme turbulence model and a mean wind speed of $9 \mathrm{~m~s^{-1}}$. Safety factors are chosen according to IEC 61400-1 (2019), but the partial safety factor for loads was increased from 1.35 to $1.35 \cdot 1.2 = 1.62$ to account for the simplified loads analysis for this preliminary optimization study.

The cost model implemented in *WISDEM* based on the work from Fingersh et al. (2006) was used to create a breakdown of the costs of major wind turbine components. The model includes a rather detailed estimation of the blade costs, as described by Bortolotti et al. (2019), including assumptions for materials, labour, tooling and many more aspects. On the contrary, the costs for parts like the pitch system and the hub are implemented as simple functions of the rotor diameter or the blade mass. The assumption of the direct drive generator costs was adjusted since the original model only takes the machine rating as an input. In our case, the rated power remains constant but the rated torque increases since the maximum rpm is reduced (constant maximum blade tip speed). According to Fingersh et al. (2006), the generator mass scales with $M_{g,rated}^{0.606}$, with $M_{g,rated}$ being the rated generator torque. We accounted for the mass increase in the cost estimation, assuming that the costs increase linear with the mass. Overall, the cost model can serve to point out trends in the development of costs when increasing the turbine size, but absolute values should be handled with care.

Aeroelastic simulations are carried out with *OpenFAST V3.1* (Jonkman and Sprague, 2021). The aerodynamic modelling includes the effects of tower shadow and the aerodynamic loading on the tower, as well as the Minemma/Pierce dynamic stall model, as described by Damiani and Hayman (2019). For the purpose of this conceptual study, the authors chose *ElastoDyn* as a structural model which uses the Euler-Bernoulli beam theory with fitted mode shapes. This module is computationally inexpensive and allows for efficient design iterations. The shortcoming is the low modelling fidelity as the blades are modelled as straight, isotropic beams and blade torsion is neglected. For high-fidelity investigations, this assumption does not hold true for blades of this size and further simulations are planned using the fully coupled 6x6 mass and stiffness matrices to elaborate the impact of the simplifications made with the low-fidelity structural model. The authors are also aware that aeroelastic stability can be a main design driver for the structural design of such a long and slender blade. However, this is out of the scope of this aerodynamically focused paper.

To investigate the aeroelastic behaviour of the *Hybrid-Lambda Rotor* under transient inflow conditions, a set of design load cases (DLCs) is defined in Table 1. The DLCs are numbered according to the standard DNV-GL ST-0437 (November

2016). From preliminary investigations, the most critical DLCs were identified as extreme wind shear, storm events with yaw misalignment and power production with normal turbulence model (NTM). Those DLCs are simulated for the *Hybrid-Lambda Rotor* and the IEA 15 MW reference turbine with the same simulation methodology with the exception of DLC 6.3, which could not be simulated for the IEA 15 MW reference turbine due to stability problems for large yaw angles.

**Table 1.** Set of ultimate design load cases investigated in aeroelastic simulations

| DLC | Description | wind speed at hub height $[\mathrm{m\ s^{-1}}]$ |
|---|---|---|
| DLC 1.5 | Transient extreme wind shear (vertical and horizontal wind shear pos. and neg.) | 10 |
| DLC 1.6 | Power Production, NTM | (5, 7.5, 10, 13, 18, 21) |
| DLC 6.1 | Extreme wind speed model, 50-years storm, yaw misalignment of +/- 8° | 50 |
| DLC 6.3 | Extreme wind speed model, 1-years storm, yaw misalignment of +/- 20° | 40 |

## 3   Results

In this chapter, we focus on the given use-case of the 15 MW offshore wind turbine, no longer generalizing the concept, in order to simplify the understanding. This means only one specific turbine diameter is presented here, although the influence of the rotor radius as a design variable was investigated and is further described below. We first address the resulting aerodynamic blade design and the influence of certain design variables. Table 2 summarizes general turbine parameters. The second part deals with loads, axial induction, angle of attack and power generation under steady and uniform inflow conditions. This is followed by the results of the structural design and the aero-servo-elastic investigations.

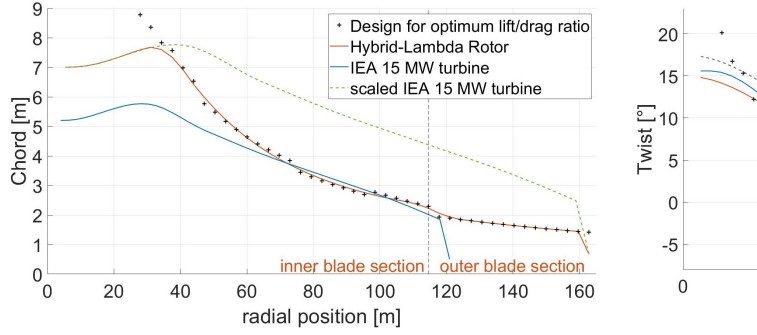

**Figure 4.** Chord distribution over radius

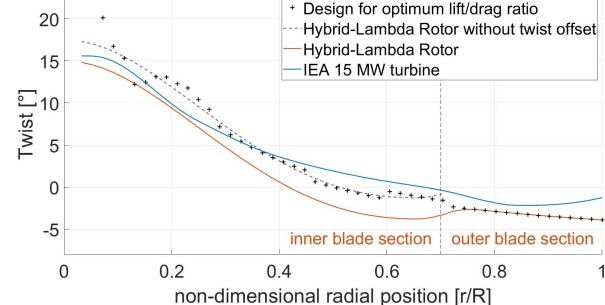

**Figure 5.** Twist over non-dimensional radius

### 3.1   Aerodynamic blade design

The *Hybrid-Lambda Rotor* has a specific rating of $180\ \mathrm{W\ m^{-2}}$ and a rotor diameter of $326\ \mathrm{m}$, which corresponds to an increase in rotor diameter and the swept area by a factor of 1.36 and 1.84, respectively. As shown in Fig. 4, the above-described design

principles lead to a very slender blade due to the design for a lower induction factor of 0.21. The outer blade section is even more slender due to the design for the higher TSR. Nevertheless, the blade root diameter is scaled by the same factor as the rotor radius to create space for a larger pitch bearings. Figure 5 shows the blade twist. Of particular interest are the negative twist offset in the inner blade section and the design for optimal AoA in the outer blade section when operated at zero pitch in light winds. The twist offset is further described and investigated in Sect. 3.2.

**Table 2.** General parameter of the *Hybrid-Lambda Rotor* and the reference turbine

| Description | Symbol | Hybrid-Lambda Rotor | IEA ref. turbine | Unit | Type of design variable |
|---|---|---|---|---|---|
| number of rotor blades | $B$ | 3 | 3 | - | fixed |
| rated power | $P_{rated}$ | 15 | 15 | MW | fixed |
| rotor diameter | $D$ | 326 | 240 | m | optimized |
| specific rating | | 180 | 332 | W m$^{-2}$ | optimized |
| hub height | | 193 | 150 | m | derived from rotor diameter |
| design TSR inner 70 % of blade span | $\lambda_{d,inner}$ | 9 | 9 | - | optimized |
| design TSR outer 30 % of blade span | $\lambda_{d,outer}$ | 11 | 9 | - | optimized |
| max. rotor speed | $\omega_{max}$ | 5.38 | 7.56 | rpm | derived from rotor design |
| min. rotor speed | $\omega_{min}$ | 2.8 | 5.0 | rpm | derived from rotor design |
| max. blade tip speed | | 95 | 95 | m s$^{-1}$ | fixed |
| cut-in wind speed | $v_{cut-in}$ | 3 | 3 | m s$^{-1}$ | fixed |
| rated wind speed | $v_{rated}$ | 10.2 | 10.6 | m s$^{-1}$ | derived from rotor design |
| cut-out wind speed | $v_{cut-out}$ | 25 | 25 | m s$^{-1}$ | fixed |
| blade prebend | | 5.43 | 4.0 | m | fixed/derived from diameter ratio |
| unloaded tip-to-tower-clearance | | 38 | 32 | m | fixed/derived from diameter ratio |
| rotor precone angle | | 4 | 4 | ° | fixed |
| shaft tilt angle | | 6 | 6 | ° | fixed |
| blade mass | | 140 | 65 | t | optimized |
| max power coefficient | $c_p$ | 0.48 | 0.489 | - | derived from rotor design |
| thrust coefficient in light wind mode | $c_{t,LW}$ | 0.78 | 0.799 | - | derived from rotor design |
| thrust coefficient in strong wind mode | $c_{t,SW}$ | $\leq 0.52$ | 0.799 | - | derived from rotor design |

In the following paragraphs, we discuss some challenges in the design process and the influence of certain design parameters. First, we discuss the influence of the specific rating and thus the rotor radius. In these considerations, we assume the distribution of axial induction unchanged, as well as the relative position of the transition point between the inner and outer blade section, which are addressed in the next paragraphs. When varying the rotor radius, we still keep the objective of limiting

the steady-inflow flapwise RBM to the maximum value of the reference turbine. If the rotor radius is enlarged, the power output is increased before the limiting loads are reached (e.g., in region 1 and 2.1). But at higher wind speeds, when peak shaving is applied (in region 2.2 and 2.3), the blade must be pitched further and power losses are more pronounced. This can even lead to the fact that the turbine reaches rated power at higher wind speeds compared to the initial design. Furthermore, at a lower wind speed, the limiting loads are reached and the blade operates in a wider range at maximum loads (flapwise RBM). The choice of the optimal rotor radius is therefore taken by carefully choosing a reasonable value for the wind speed at which the limiting loads are reached (in this case about 70% of rated wind speed) and with the aim of reducing the rated wind speed compared to the reference turbine. In the given case, this leads to a specific rating of $180 \text{ W m}^{-2}$ and a diameter of 326 m for the conceptual design. Typical specific ratings for offshore turbines are in the order of magnitude of $400 \text{ W m}^{-2}$ and are expected by Baumgärtner et al. (2021) to drop to approximately $330 \text{ W m}^{-2}$ in the next years. For onshore turbines Baumgärtner et al. (2021) reported a drop in the average specific rating for newly installed turbines in Germany from around $400 \text{ W m}^{-2}$ in 2012 to $300 \text{ W m}^{-2}$ in 2018. These numbers highlight the extremity of the *Hybrid-Lambda Rotor* design and clearly show the change in design philosophy.

As a second aspect, we address the relative position of the transition point between the inner and outer blade sections. The induction distribution within the two sections is again considered unchanged. In this case, a shift of the transition point towards the blade tip increases the section of the blade, that experiences higher loading. Therefore, the turbine reaches the limiting loads at lower wind speeds and the range of peak shaving gets wider. Again, this shifts the compromise finding towards power maximization in the light wind mode. If the transition point is moved too far towards the tip, the tip loss effects can overshadow the beneficial aerodynamic influence of the low loaded tip section. In the present study, we chose the transition point to be at 70% blade span, which allows for two descriptive explanations: First, the swept area of the two blade sections have almost identical sizes (inner rotor disk and outer rotor annulus). Second, the inner rotor disk has almost the same size as the swept area of the reference turbine, which means the slender outer section of the blade can be interpreted as an extension of the original blade.

As a third design variable, the axial induction is addressed. When a blade operates at a higher TSR than it is designed for, an undesired increase in axial induction along the blade span towards the tip is observed. This means, when plotting the axial induction vs. radius, the line tilts upwards. The design point for the inner blade section is the low TSR of 9 and an axial induction factor of 0.21 with a pitch angle of 2.2° to achieve the above-mentioned change in AoA distribution. At the same time, when operated at high TSR and zero pitch, an axial induction factor of 0.33 is desired with a distribution that is as uniform as possible over the inner blade section. To find a compromise between these two requirements, the induction in Eq. (1), Eq. (2) and Eq. (4) is linearly lowered from 0.225 at 30% blade span to 0.17 at the end of the inner blade section (at 70% blade span) before the twist offset is applied. In this way, we can achieve a fairly constant axial induction in the inner section in the light wind mode, while in the strong wind mode a modest decrease of the axial induction towards the tip is beneficial for load reduction. Figure 6 shows the axial induction for various operational points as a result from stationary BEM simulations. In

fact, the decreasing axial induction towards the blade tip in the strong wind mode (see Fig. 6, yellow line) shows a similar trend as derived by Jamieson (2020), which is beneficial for rated load reduction.

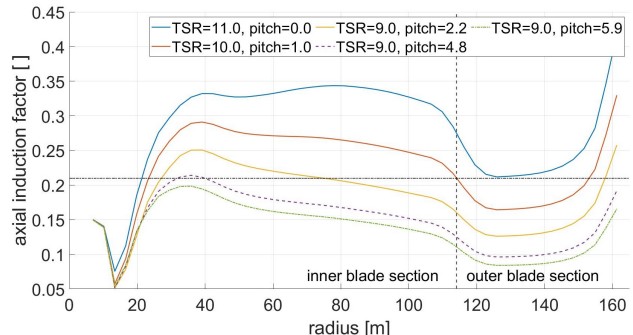

**Figure 6.** Axial induction: Light wind mode (blue), transition (red), strong wind mode (yellow), peak shaving (purple, dashed), rated power (green, dash-dot), design value 0.21 (black, dash-dot)

## 3.2 Aerodynamics, loads and power under steady-inflow BEM simulations

After addressing the blade design, we present results from the BEM simulations and the derived control strategies with steady and uniform inflow and with rigid structures. The concept is further compared to the reference turbine. Note, that due to the gradients along the blade span the assumptions made in the BEM theory can reach their limit. We used free-vortex wake methods to investigate to what extend the assumption of independent blade elements in the BEM theory is violated. Results show good agreements for rotor integrated quantities (power and thrust), although some differences are noticeable in the radius resolved variables when the gradients along the blade span are large in the light wind mode. The interested reader is referred to Ribnitzky et al. (2023).

First, the control strategy for the *Hybrid-Lambda Rotor* is presented in Fig. 2. At a cut-in wind speed of $3 \, \mathrm{m \, s^{-1}}$ and above, the minimum rotational speed is set to $2.78 \, \mathrm{rpm}$, which is approximately half of the maximum rotational speed. From $4 \, \mathrm{m \, s^{-1}}$ on, the rotor operates at the high TSR of 11 in the light wind mode and a fine pitch angle of $-0.8°$ which leads to the maximum power coefficient. This pitch angle is called fine pitch since the pitch angle for optimal $c_p$ was derived after the blade design was concluded. The limiting loads are reached at $6.8 \, \mathrm{m \, s^{-1}}$. For higher wind speeds, peak shaving ensures the limitation of the flapwise RBM and the rotational speed is kept constant until the lower TSR of 9 is reached. During the strong wind operation mode, this TSR is maintained until the maximum rotor speed is approached. In the pitch schedule in Fig. 2 it is visible that the necessary pitch angle to limit the flapwise RBM increases slower in region 2.2 (where the transition of the operating modes takes place) compared to region 2.3. From a wind speed of $15 \, \mathrm{m \, s^{-1}}$ rated power and the rotational speed are linearly decreased. This takes into account the oversupply of wind power from conventional turbines and it is expected to further reduce the fatigue loads of the very slender blades.

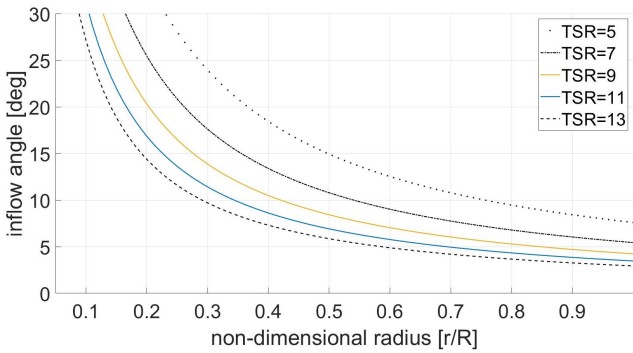

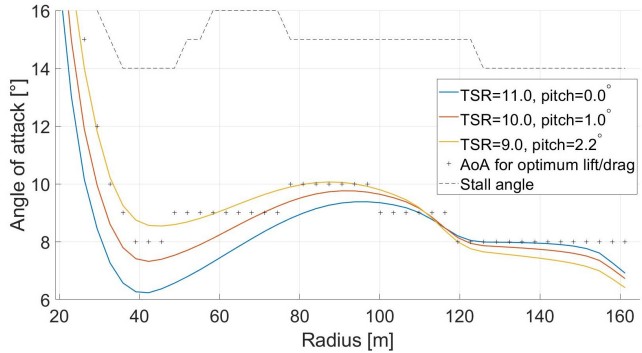

**Figure 7.** Inflow angle distribution for varying TSR (assuming constant $a = 1/3$ over the blade length)

**Figure 8.** Angle of attack (BEM): Light wind mode (blue), transition to strong wind mode (red), strong wind mode (yellow)

To better understand the aerodynamic concept and to take additional benefits from the operation with two design TSRs, we take a closer look at the transition between the two design points. When the TSR is lowered (constant rpm, increasing wind speed), the AoA and therefore the lift coefficient increases because the axial flow component increases and the circumferential component remains constant. But in terms of reducing the loads, an increase in AoA is not desired. For an ideal transition from low wind to strong wind mode, the blade would see an increase in AoA in the inner section and a decrease in AoA in the outer section. This would shift the power generation to the inner section and reduce the lever arm of the resulting bending forces. Figure 7 shows the inflow angle distribution over non-dimensional radius for varying TSR for a certain constant axial induction factor a, as described in Eq. (5) according to Gasch and Twele (2012). We plotted a wider range of TSR to emphasise the differences.

$$\phi(r) = \arctan\left( (1 - a)\frac{R}{r\lambda} \right) \tag{5}$$

This formula is valid for a constant axial induction factor along the blade span which is not the case for the given concept. But to simplify the understanding, we will use the formula for the following findings. Obviously, closer to the root the change in the inflow angle is greater than at the tip. Thus, the above-mentioned change in AoA distribution (i.e. inboard increase and outboard decrease) can be achieved if the blade is pitched towards feather simultaneously while reducing the TSR. To the authors' knowledge, this technique has never been documented before in wind energy applications and we will use this technique to fine tune the aerodynamic behaviour during the transition between the two operational modes, emphasising the change in AoA. Figure 8 shows the angle of attack distribution for the *Hybrid-Lambda blade*. The outer blade section operates at the optimal AoA in the light wind mode, while the inner blade section operates below the optimal AoA because it is operated at a higher TSR than it was designed for. When transitioning to the strong wind mode (i.e. lowering the TSR and pitching to feather), the AoA decreases in the outer section but increases in the inner section. In this way, the inner section operates closer

to the optimal AoA and the change in the AoA distribution reduces the lever arm of bending forces.

In this paragraph, we analyze the power output of the *Hybrid-Lambda Rotor* and investigate whether the desired advantages are met. Close to cut-in wind speed, wind turbines operate at very high TSR because the minimum rotational speed is set by the generator characteristics and by the first tower eigenfrequency. As seen in the power curve, the presented concept (solid red line in Fig. 9) shows advantages because the light wind design point of the blade is at a higher TSR and therefore closer to the operational TSR at cut-in. The *Hybrid-Lambda Rotor* operates at the high design TSR up to $6.8 \mathrm{\ m \ s^{-1}}$ until the limiting flapwise RBM is reached. At this point, the turbine's power is 1.8 times greater than the reference turbine's power (blue line). The aerodynamic power coefficient in the light wind mode is 0.481, which is only 1.7% lower than the maximum power coefficient of the reference turbine. At higher wind speeds, the turbine operates at the lower TSR and peak shaving through pitching to feather is applied to limit the loads. Here, the different efficiencies in peak shaving are visible. The potential is shown by the red dashed line, which represents the power output if the loads would not be limited through peak shaving. The green dashed line indicates the power curve of the reference blade that is geometrically scaled by the same factor and conventional peak shaving is applied to limit the flapwise RBM. This means only the pitch angle is set to a higher value to constrain the flapwise RBM while the rpm follows the design TSR. In contrast, the black dotted line represents the same blade (geometrically scaled IEA 15 MW) but peak shaving is applied in a similar manner as for the *Hybrid-Lambda Rotor*. This means for $v > v_{shift,start}$ the rpm is kept constant until the operational TSR is reduced from 9 to 7. For $v > v_{shift,end}$ the rpm schedule follows the TSR of 7 which is an arbitrary choice in this case and should be optimized in a detailed design study. In addition, the pitch angle is set for $v > v_{shift,start}$ in order to limit the flapwise RBM. In short, we are applying the *Hybrid-Lambda* control strategy to a conventional blade design. The results show that the power output can be greatly increased if the TSR is lowered in region 2.2 and 2.3 (compare green dashed and black dotted line in Fig. 9). Thus, peak shaving should not only be accomplished by increasing the pitch angle, but also by optimizing the operational TSR with respect to the load constraint (as also indicated by Madsen et al. (2020)). Since the results show that a reduction of the operational TSR is beneficial in the peak shaving region it makes sense to account for this fact already in the blade design which is integrated in the *Hybrid-Lambda* design methodology. Indeed, the *Hybrid-Lambda Rotor* enables even lower power losses in the peak shaving region since the TSR reduction is already accounted for in the blade design (compare solid red and dotted black line in Fig. 9). The turbine concept reaches its rated power at $10.2 \mathrm{\ m \ s^{-1}}$, which is $0.4 \mathrm{\ m \ s^{-1}}$ lower than the reference turbine.

Further benefits become clear when looking at the thrust characteristics of the *Hybrid-Lambda Rotor* in Fig. 10. Low thrust coefficients are beneficial in terms of design driving loads and wind farm efficiency. The maximum thrust coefficient of the conceptual rotor is comparable to the reference turbine. But the real advantage lies in the transition to the strong wind mode. The blade is pitched and the thrust coefficient decreases rapidly, which leads to much lower wake losses for wind speeds greater than $6.8 \mathrm{\ m \ s^{-1}}$. The wake losses of the *Hybrid-Lambda Rotor* are addressed by Ribnitzky et al. (2023). Results show significant advantages even in a scenario with constant absolute spacing (compared to the IEA 15 MW reference turbine). The maximum dimensional steady thrust is only 3% higher than the maximum value for the reference turbine. Since the thrust is

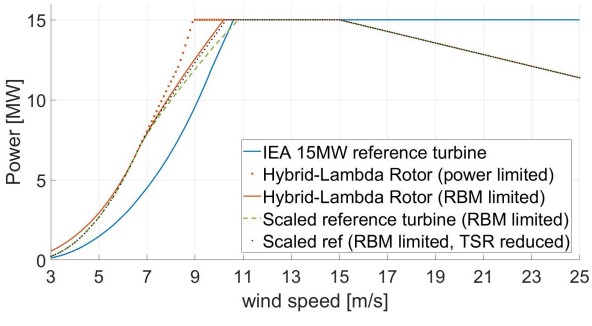
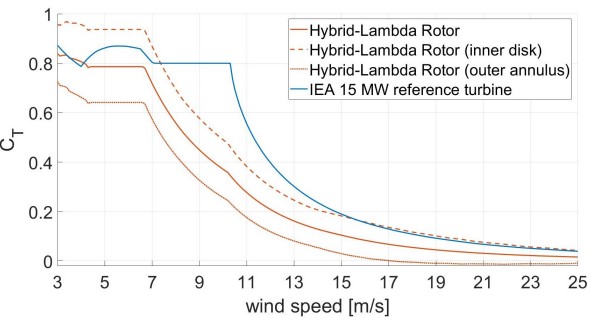

**Figure 9.** Power output of the *Hybrid-Lambda Rotor* (solid red) compared to the reference turbine (blue) and a scaled reference turbine (dashed green and dotted black)

**Figure 10.** Thrust coefficient of the *Hybrid-Lambda Rotor* (RBM limited) compared to the reference turbine

employed over the 85% larger swept area, the thrust coefficient drops significantly. A breakdown of the contribution to the thrust coefficient of the inner and outer parts of the rotor is depicted in Fig. 10. Those are calculated by separately considering the respective swept area and blade span for the actual thrust force and reference force. As the inner rotor operates at higher axial induction factors (in light wind mode) compared to the reference turbine, higher thrust coefficients are derived. But for wind speeds greater than $7.4 \, \mathrm{m \, s^{-1}}$ the thrust coefficient of the inner rotor disk is lower than the value of the reference turbine due to peak shaving.

### 3.3 Optimization of the structural blade and tower design

Multi-disciplinary design and optimization routines are nowadays common practices in wind energy research. Consequently, the objective of this section is not to provide a new design methodology but to elaborate that a consistent and realistic system design is also possible for such an innovative and in some sense unconventional rotor design. A common problem with very long and slender blades is the low area moment of inertia that such cross-sections provide. Thus, the stiffness provided by the geometric shape is relatively low and massive reinforcements by very thick carbon spar caps need to be added. This often leads to heavy and expensive blades, which means that the advantage of a slender blade, that is expected to use less material, is counterbalanced. This thesis is supported, as the initial blade design of the *Hybrid-Lambda Rotor* is only 5% lighter than expected by the cubic scaling law if the reference blade would be scaled geometrically (see Fig. 11).

A possible solution is to increase the relative thickness (which is equivalent to pushing the thicker airfoil sections closer to the tip) to increase the geometric area moment of inertia and reduce the spar cap thickness accordingly to save material, mass and costs. On the one hand, the designer sacrifices aerodynamic efficiency and consequently AEP, as the thicker airfoils are less efficient. On the other hand, the blade can be designed lighter and less costly. This trade-off is made by minimizing the cost of valued energy (COVE), using the multi-disciplinary optimization algorithm *WISDEM* which was adjusted by the authors to

implement the *Hybrid-Lambda* design methodology, as described in Sect. 2.3. Three design parameters are compared along the blade span between the initial and the optimized blade in Fig. 11. The relative thickness for the inner blade section is increased while the airfoils are locked for the outer blade section in order to maintain the aerodynamic characteristics of the tip region, which is designed for the light wind regime. The spar cap thickness is reduced, especially for the root section ($0 < r/R < 0.4$). This section sees much larger chord lengths, as the blade root diameter is scaled by the same factor as the increase in rotor diameter to create space for larger pitch bearings which will be needed to carry the increased edgewise bending moments. Due to this large chord length close to the root and the further increase of relative thickness in the optimization, the area moment of inertia is relatively large and the spar cap thickness can be reduced significantly. This leads to a more uniform distribution of the strains in the spar caps, as shown in Fig. 11c.

Special care needs to be taken on the transition of the inner and the outer blade section at 70% blade length. In this area, the chord length as well as the aerodynamic forces change over a small span-wise section. In the initial design, this area was reinforced by a thicker spar cap layup (see dashed blue line in Fig. 11), to reduce the peak that was observed in the strains in this transition area. Of course, such a distribution of spar cap thickness is not beneficial in terms of eigenfrequencies because it adds more mass closer to the tip, which drives down the eigenfrequencies and most likely has to be compensated by a stiffer design of the inner blade section. However, the optimization algorithm reduced the spar cap thickness in this transition area at 70% blade length, too, which did not lead to a significant increase in the strains. The resulting mass and stiffness distributions are compared to those of the IEA 15 MW in Fig. 13, clearly showing the steeper gradient in the flapwise stiffness in the transition area of the *Hybrid-Lambda* blade. The reader should bear in mind that the structural solver *PreComp* is a 2D cross sectional solver and does not account for stress concentration due to rapid changes in the geometry in span wise direction. Overall, the optimized blade is 14% lighter compared to a geometrically scaled blade of the same size. If we define $n$ as the ratio of diameters of the reference and the *Hybrid-Lambda* turbine the blade mass would increase with $n^3$, according to Gasch and Twele (2012), while our design leads to an exponent of 2.5 instead. Note, that the reference exponent of 3 is only derived by geometric considerations. Griffith and Richards (2014) summarize recent trends for commercial and research blades and state mass scaling exponents of 2.5 for moderately innovative blades and 2.1 for highly innovative designs.

The initial tower is designed as an isotropic, tapered steel tube. The design is similar to the tower of the IEA 15 MW reference turbine with the adjusted hub height and an increase of tower diameter from 10 to 11 m at the tower base. After the optimization the tower base diameter is reduced to 8.54 m and the material thickness is increased accordingly to meet the requirements for buckling, stress constraints and eigenfrequencies. The monopile diameter is reduced to the same value. Overall, the combined structural mass of tower and monopile is reduced by 17.5% compared to the initial design, which leads to significant cost savings, as further described in Sect. 3.5. As shown in Fig. 12, the tower design results in a classic soft-stiff design with the first eigenfrequencies between the rotational (1P) and blade passing (3P) frequency. The first flapwise eigenfrequency of the blades is safely above 3P excitation.

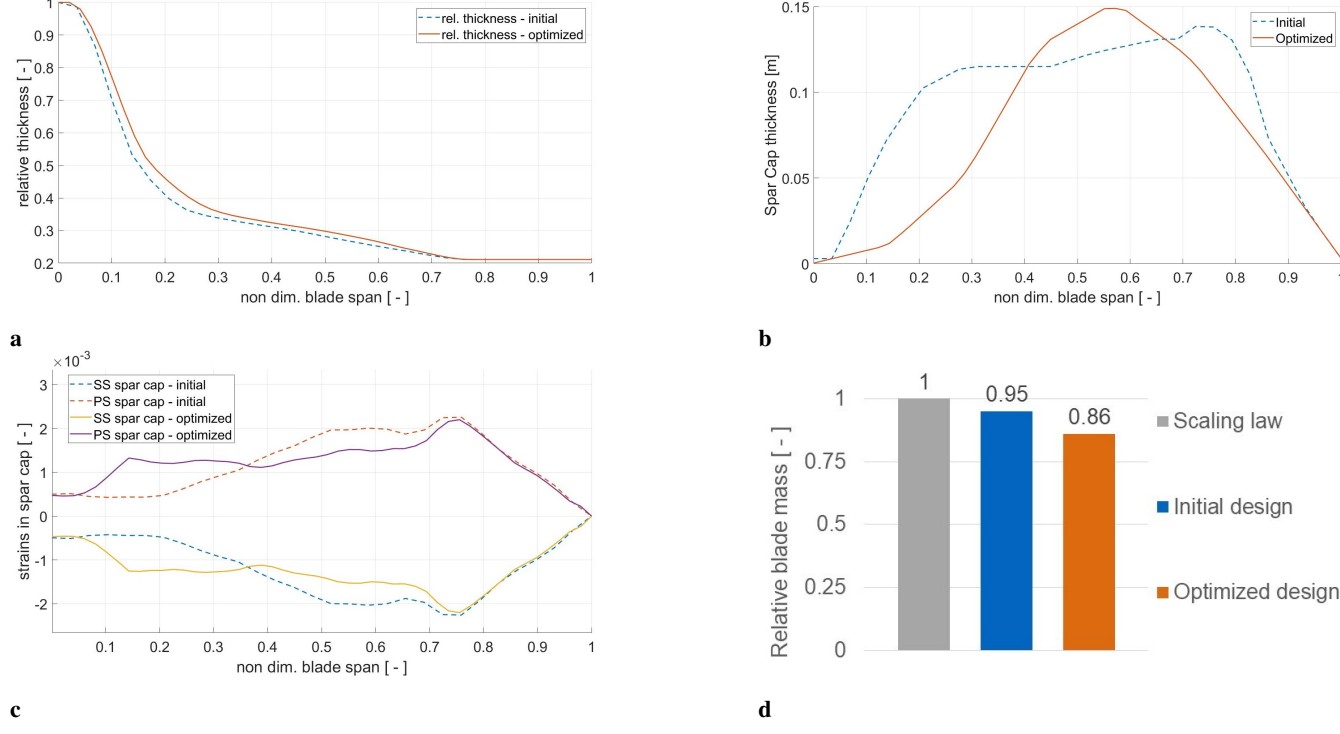

**Figure 11.** Comparison of structural parameters between initial (dashed lines) and optimized (solid lines) blade design: Relative thickness distribution (a), Spar cap thickness distribution (b), Strains in the spar caps (SS = suction side, PS = pressure side) (c), blade mass relative to cubic scaling law of IEA 15 MW blade mass (d)

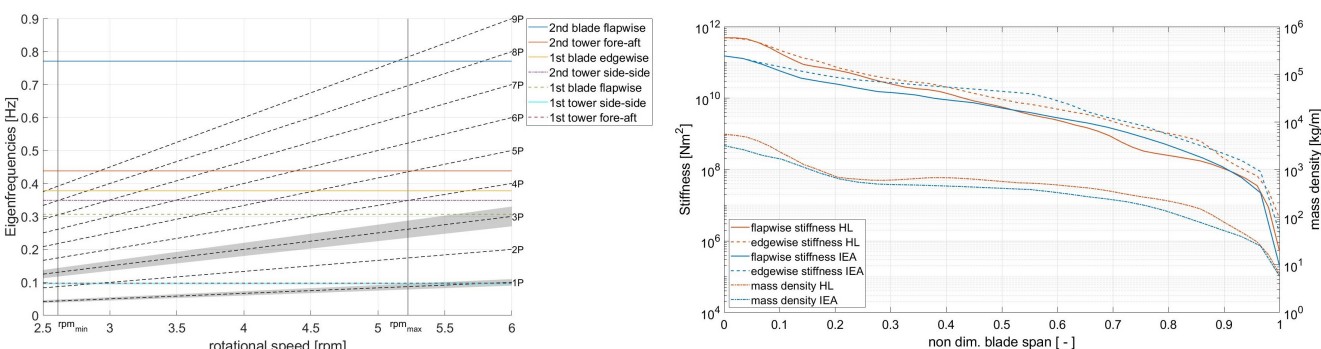

**Figure 12.** Campbell diagram for optimized blade and tower design. 10% safety margins for 1P and 3P excitations shown in grey.

**Figure 13.** Mass and stiffness distribution for the optimized *Hybrid-Lambda* blade (red) and the IEA 15 MW (blue)

## 3.4 Aeroelastic load simulations

After setting up a consistent turbine model, we investigate the *Hybrid-Lambda Rotor* with aeroelastic simulations. First, the controller design is described. In section 3.4.2, we describe the transition between the operating modes in a turbulent wind field. We further investigate a set of design load cases in section 3.4.3 and in section 3.4.4 we examine extreme wind shear events. The presented conceptual investigation does not include a fatigue analysis. Despite the general importance, this is considered out of the scope of the paper, which is further discussed in Sect. 4.

### 3.4.1 Initial controller design

The innovative operating characteristics described in Sect. 3.2 open new challenges and opportunities in the design of the controller. The partial load regime below rated is no longer governed by a pure torque controller. Two newly arising challenges are the transition between the operating TSRs and the limitation of the flapwise RBM. Changing the TSR is certainly an important topic for the controller design and drive train dynamics, given an increase of the rotational inertia of the *Hybrid-*
460 *Lambda Rotor* compared to the reference turbine by a factor of 3.5. According to generic scaling laws, the rotational inertia increases with $n^5$, while the slender blades and lower radius of the centre of mass of the *Hybrid-Lambda Rotor* lead to an exponent of 4.1 instead. In this study, we aim for a controller that fulfils these basic requirements by using only standard methods like pitch and torque control. Of course, an advanced controller design like individual pitch or observer-based feed-forward control would be beneficial for a rotor of this size, but it is not in the scope of this paper.

To better describe the control strategy, we will distinguish four regions below rated power, as depicted in Fig. 2. Following a certain TSR (11 in region 2.1, or 9 in region 2.3) is achieved by setting the generator torque as

$$M_{\mathrm{g}} = \frac{\pi \, R^5 \, \rho \, c_p(\omega)}{2 \, \lambda^3} \, \omega^2 \tag{6}$$

Note, that there is no unique $c_p$ in region 2.3 since the pitch angle is a function of wind speed. Hence, the desired $c_p$ from steady-inflow simulations is implemented as a function of rotational speed. The transition between the two TSR values is based
on a PI controller, similarly to how the transition to the full load region is realized or how the torque control near the cut-in wind speed is done, as described by Burton et al. (2011). During the transition between the two TSR values (region 2.2), the PI controller keeps the rotor speed at the constant value $\omega_{\mathrm{shift}}$ by setting the generator torque. To assure a smooth transition to the constant-TSR regions, the PI output is constrained by the torque values from Eq. (6) corresponding to $\mathrm{TSR} = 9$ and $\mathrm{TSR} = 11$, respectively. In that way, the PI-torque controller is active only during the transition region, while the constant TSR
regions are achieved by the torque control law from Eq. (6), similar to conventional turbines in partial load. A similar PI-based solution is also used for conventional turbines to avoid too low and too high rotor speed values near the cut-in and the rated wind speed, as described in Burton et al. (2011).

For the pitch controller two versions are implemented. The first version is referred to as simplified controller and implements
the transition of the TSR and a look up table for the pitch signal for regions 2.2 and 2.3. This simplified controller is used for the

load case calculations in Sect. 3.4.3. A second version is developed that features a feedback from the flapwise RBMs, further referred to as load feedback controller and it is applied in Sect. 3.4.2. The functionality of the pitch controller can be described in three parts. Firstly below rated, we use a conventional implementation with a look-up table for region 1 and a constant pitch for the maximum power output in region 2.1. The argument of the look-up table is the filtered wind speed, mimicking a wind speed estimator. Secondly, above rated, a standard PI-pitch controller ensures a constant rotational speed or the linear decrease of rotational speed in region 3.2. Thirdly, in parallel to these two functionalities, we implemented a load limiter (for region 2.2 and 2.3). To do so, the mean of the three flapwise RBMs is low-pass filtered and fed back to the controller. The RBM feedback is then compared to the maximum allowed flapwise RBM. As long as the RBM feedback is larger than the constraint, the reference pitch value (output of the controller) is increased, thus not allowing the blades to reduce its pitch angles, which would further increase the RBMs. The change of the reference pitch angle is proportional to the difference between the RBM feedback and the constraint. The output signal is saturated with the maximum pitch rate of $3°s^{-1}$. The proportional gain is a controller parameter that needs to be tuned. Once the RBMs drop below the constraint, the reference pitch angle is also reduced, thus allowing the PI-pitch controller to reduce the pitch angles to the optimum pitch in region 2 or to keep the rotor speed constant in region 3.1. In this way, the amplitude of load variations can be drastically reduced and load overshoots are less severe. Nevertheless, the increased pitch activity needs to be considered when sizing the actuators and bearings which will influence the resulting cost function. Instead of using the mean of all three RBMs it's also possible to use the maximum of the three signals as a load feedback. The implementation would be identical as described above. This results in larger pitch angles and lower loads but obviously also reduces the energy yield. The choice of the respective feedback signal and the magnitude of the constrained RBM is always a compromise finding between load limitation and power maximization. In an advanced setting of turbine control and structural health monitoring, this could also be adjusted throughout the lifetime of the turbine, considering the actual condition of the turbine.

Additionally, advanced control designs are required to further limit the loads in dynamic and turbulent inflow conditions. In fact, some findings from the aeroelastic simulations show a need for an improved controller design, but this is in the scope of future work.

### 3.4.2 Transition of operating modes in a turbulent wind field

From a conceptual viewpoint, the ability of the *Hybrid-Lambda Rotor* to switch between the light wind and strong wind operating mode under turbulent inflow is of special interest. For this purpose, a wind field with normal turbulence model, a mean wind speed of $7.5 \mathrm{~m~s^{-1}}$ and a turbulence intensity of $21\%$ is chosen and the ability of the load feedback controller is tested. The results are shown in Fig. 14 (TSR, wind speed and flapwise RBM are low-pass filtered). Considering steady inflow, the maximum flapwise RBM (dashed black line) is reached at a wind speed of $6.9 \mathrm{~m~s^{-1}}$ ($v_{shift,start}$ – dashed blue line) and a pitch angle of $0°$, and according to the design concept the rotor should switch to the strong wind mode (lower TSR) for higher wind speeds. The time periods with the wind speed at hub height bellow $v_{shift,start}$ are marked with a green background to indicate that the rotor should be (in theory) in the light wind mode with the high TSR. For wind speeds greater than $8.3 \mathrm{~m~s^{-1}}$ ($v_{shift,end}$ - dashed-dotted blue line) the rotor should operate at the low TSR and those time periods are marked in red. The

515 time periods with the wind speed between $v_{shift,start}$ and $v_{shift,end}$ are marked in yellow, indicating that the rotor is transitioning between the two operating modes. Minor short-term exceedances of the transition wind speed are not highlighted.

In Fig. 14, three transitions from light wind to strong wind mode are visible. The first one starting at 212 seconds, shows a sudden gust event with an increase of wind speed from $5.5$ to $9.2\,\mathrm{m\,s^{-1}}$ in only 14 seconds. The rotor is able to reduce the TSR

from 11 to 9.75 in 15 seconds. The attentive reader will notice that for large rotors with a high moment of rotational inertia, the TSR will decrease by default for a sudden increase in wind speed, as the rotor speed changes much slower than the wind speed. However, as a reaction to this gust event the controller increases the torque rapidly (as shown with the dashed-dotted red line) to keep the rotational speed constant and to even enforce the reduction of the TSR. The second ramp event takes place at 275 seconds and the controller reaction is similar to the above-described case. The third transition to the strong wind mode

occurs at 390 seconds and for the following relatively long strong wind period, the controller is able to reduce the TSR fully to the desired value of nine. However, this transition is rather slow, occurring over a time period of 37 seconds. The transitions from strong wind back to light wind mode take place at 175, 227 and 335 seconds. Especially the latter transition happening relatively quickly (12 seconds from TSR 10 to 11), reaching the desired TSR of 11 exactly at the time when the wind speed drops below $v_{shift,start}$.

This investigation shows that it is in general possible to apply the control strategies described in Sect. 3.2 in a turbulent wind field in a fully aeroelastic simulation. Although the transition between the operating modes is found to be possible, this method is more suitable for slow changes in the mean wind speed. Sudden gust events, as observed at 212 seconds, still lead to a minor exceedance of the flapwise RBM compared to the limit that was applied in steady BEM calculations. Furthermore, it becomes

clear that the current controller setup could imply large torque variations and associated high drive train fatigue loads. More advanced controller concepts (e.g. model predictive control) are likely required.

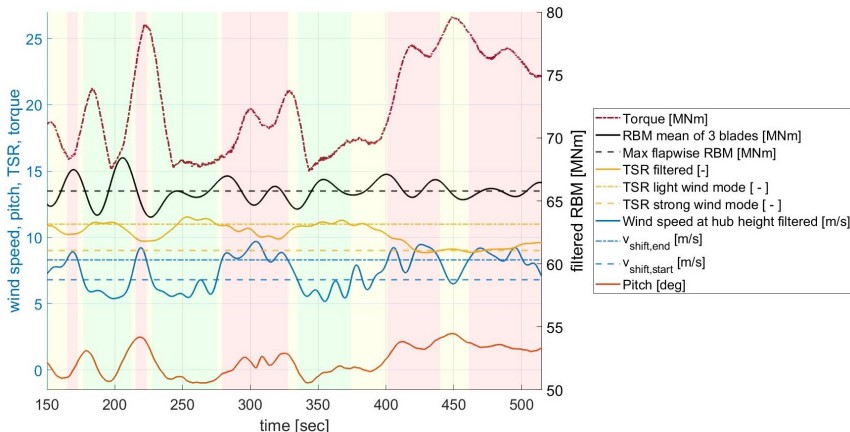

**Figure 14.** Transition of operating modes in a wind field with normal turbulence model and a mean wind speed of $7.5\,\mathrm{m\,s^{-1}}$, background colours indicate the desired operating modes, green: Light wind, red: Strong wind, yellow: Transitioning between the operating modes.

### 3.4.3 Comparison of ultimate loads with the reference turbine

After investigating the transition between the operating modes, we address ultimate loads in the set of design load cases from Table 1. To reveal further advantages or disadvantages of the aerodynamic design of the *Hybrid-Lambda Rotor* we performed these simulations with the simplified controller without the feedback of the loads (only implementing the transition of the TSR with a look up table for the pitch). Figure 15 presents the ultimate loads of the *Hybrid-Lambda Rotor* with solid bars and those from the reference turbine with hatched bars. Three groups are distinguished by their texture. First, the white bars illustrate the maximum loads under steady and uniform inflow including elastic deformations. Two wind speeds (rated and $v_{shift,start}$) were investigated and the more severe case is displayed here. Second, the grey bars show the theoretical load increase according to the generic scaling law as described by Gasch and Twele (2012), which would apply to a geometrically scaled reference turbine without changing the aerodynamic concept (e.g. scaling the steady-inflow loads of the IEA 15 MW, displayed with white hatched bars). Meaning that flapwise RBM scale with $n^3$, edgewise mass-driven RBM with $n^4$, thrust with $n^2$ and tower base fore-aft bending moment with $n^2 \cdot n_{tower}$ (neglecting the rotor mass increase), where $n$ is the scaling factor of the rotor diameter and $n_{tower}$ the scaling factor of the hub height. The out-of-plane tip deflection scales with $n$ assuming that the aerodynamic forces scale with $n^2$, geometrical dimensions scale with $n$ and the second area moment of inertia of the blade cross-section scales with $n^4$. The unloaded tip-to-tower-clearance scales with $n$, too (neglecting gravitational effects). Thus, the loaded tip-to-tower clearance scales with $n$ as it is the difference of two variables, both scaling with $n$ (the unloaded tip-to-tower clearance and the maximum tip deflection with the blade in front of the tower). These scaling factors are only an indication for the upper bound since the design methodology of the *Hybrid-Lambda Rotor* includes peak shaving with a constant flapwise RBM. Third, the coloured columns relate to the dynamic load quantities from aero-servo-elastic simulations.

We first address the differences between the generic scaling law and the steady-inflow loads (white and grey bars). As per definition of the design methodology, the maximum flapwise RBM moment is maintained in steady and uniform inflow conditions. Nevertheless, the blade length is enlarged and the edgewise mass-driven RBM has to increase. Due to the very slender and relatively light outer blade section, the lever arm of the centre of mass is shorter in relation to the total blade length. Hence, the edgewise RBM enlarges by a factor of 2.36 (cf blade mass increased by a factor of 2.15). This means the edgewise RBM scales only with $n^{2.8}$ rather than with $n^4$ as for geometrical scaling. The tip-to-tower-clearance represents a reserve, thus a higher value indicates a safer design. Note, that the unloaded tip-to-tower clearance also increased as documented in Table 2. The loaded tip-to-tower-clearance is larger for the *Hybrid-Lambda Rotor* in steady-uniform inflow as expected by the scaling law. The thrust is expected to be lower for the *Hybrid-Lambda Rotor* as a part of the blade is designed for low induction. The tower base fore-aft bending moment is approximately the same for the two turbines in steady-inflow simulations. On the one hand, a constant thrust and an increased tower length would increase the tower base bending moment. On the other hand, the increased rotor-nacelle-assembly mass introduces a counterbalancing moment. In total, this leads to an equal tower base fore-aft bending moment in steady-inflow simulations for the two turbines.

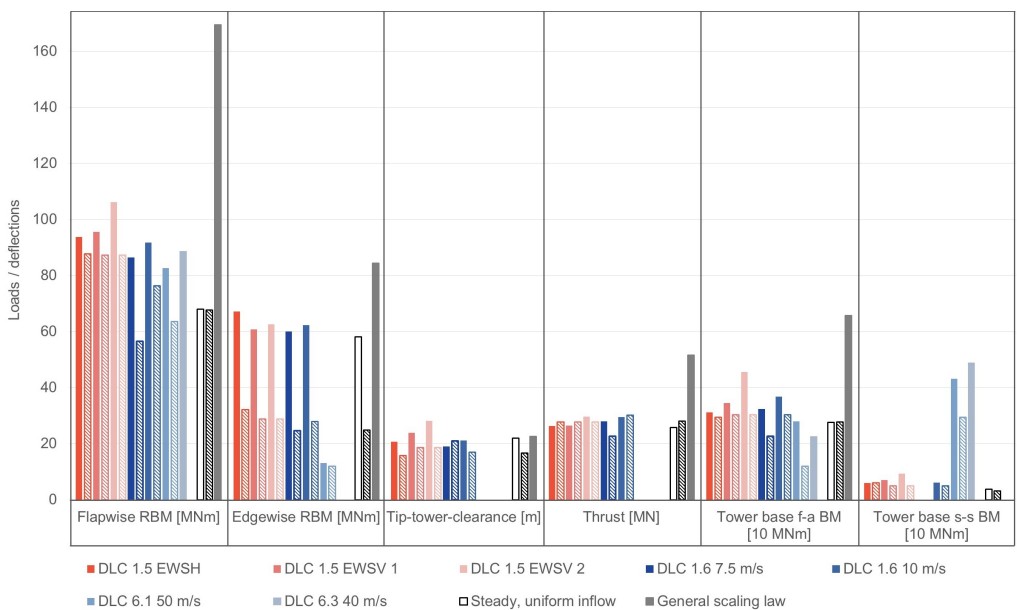

**Figure 15.** Ultimate loads in solid bars for the *Hybrid-Lambda Rotor* and in hatched bars for the IEA 15 MW reference turbine, only critical loads are displayed, EWSH = extreme wind shear horizontal, EWSV = extreme wind shear vertical, f-a BM = fore-aft bending moment, s-s BM = side-side bending moment

Next, we discuss the ultimate loads from the aero-servo-elastic simulations (coloured bars). For each type of loading, the most critical load cases are displayed as bars in Fig. 15. If a load case is not displayed for a certain type of loading, it is considered uncritical. The objective of the *Hybrid-Lambda Rotor* is to limit the stationary flapwise RBM to the maximum value of the reference turbine in steady-inflow BEM simulations. Thus, it is of special interest how much this type of loading increases in transient aeroelastic simulations. The ultimate load from normal power production is indeed marginally increased compared to the load level of the reference turbine from normal power production. But, if compared to the load level of the reference turbine under extreme wind shear events, the increase is only marginal. For the storm events, the ultimate loads could be reduced by pitching to $94°$ instead of $90°$. Still, the increase in DLC 6.3 is significant compared to the reference turbine. Due to the large yaw error of $+/-20°$ the blade experiences relatively large angles of attack (for certain azimuthal positions), which leads to an increase of 40% in the flapwise RBM. Still, for the *Hybrid-Lambda Rotor* the absolute values of the flapwise RBM for the storm events are lower than the maximum values for DLC 1.5 and DLC 1.6, which are the design driving load cases for the flapwise RBM. In the storm events, the slender blade design shows additional benefits. The shorter chord length reduces the lift forces arising from the complex interaction of blade twist, azimuthal position and yaw error.

The strongest increase in the ultimate edgewise RBM is observed for normal power production, extreme wind shear horizontal (EWSH) and extreme wind shear vertical (EWSV, the difference between EWSV 1 and EWSV 2 is explained in Sect. 3.4.4) which are the design driving load cases for both turbines. The increase is expected as much larger blades are necessarily

heavier and have a longer lever arm. But, if those values are compared to the generic scaling law, the advantage of the very slender and relatively light blades of the *Hybrid-Lambda Rotor* becomes clear. The maximum edgewise RBM is almost twice as large as for the reference turbine, but it is still only 80% of the increase expected by the generic scaling law.

The tip-to-tower-clearance turns out to be uncritical, since it is larger compared to the reference turbine. Looking at the ultimate loads for the thrust, this type of loading is uncritical, too. Since a part of the rotor is designed as a low induction rotor and the limitation of the flapwise RBM is the stronger constraint in setting up the control schedule, the ultimate thrust is lower compared to the reference turbine for this reduced set of DLCs. The tower base fore-aft bending moment is increased for the *Hybrid-Lambda Rotor* in the dynamic load cases although it is constant for the steady-inflow cases which highlights the importance of investigating transient effects.

### 3.4.4 Extreme vertical wind shear

As seen in the previous section, the *Hybrid-Lambda Rotor* is very sensitive to extreme wind shear, which we study in more detail. In the standard IEC 61400-1 (2019), transient extreme vertical wind shear is modelled in a way that the wind speed stays constant at hub height (here only shown for rated wind speed), the wind speed at the upper end of the rotor disc increases while the wind speed at the bottom of the rotor disc decreases. The unsteady event starts at 200 seconds and lasts for 12 seconds with a maximum wind speed at the top of the rotor disc after 6 seconds. We further define wind shear (s) as the slope of the wind speed (v) over the height (z), as described in Eq. 7.

$$s = \frac{\Delta v}{\Delta z} \tag{7}$$

We investigated two transient non-turbulent wind fields with different vertical wind shear. The first one (EWSV 1) implements the transient wind profiles as described in the standard IEC 61400-1 (2019) for the two turbines with the respective rated wind speed and hub height. These wind profiles are shown in Fig. 16, which makes clear that both turbines experience almost the same maximum wind speed at the top of the rotor disc and the wind speed at hub height remains constant at rated wind speed. In fact, the wind speed at the top increases from around $11$ to $18 \, \mathrm{m \, s^{-1}}$ in 6 seconds. As these wind profiles are modelled with the respective turbine parameters, this approach corresponds to a turbine type-specific wind field. Consequently, the maximum wind shear after 6 seconds is lower for the *Hybrid-Lambda Rotor* than for the reference turbine (compare dashed lines in Fig. 16), but the wind speed at the top of the rotor disk is similar. This approach is according to the standard IEC 61400-1 (2019), but it was issued for much smaller rotors. It neglects that larger rotors cover a greater spatial area and are more prone to larger differences in the spatially distributed wind speed. Therefore, we also investigated a more conservative approach. In the second case (EWSV 2), we place the *Hybrid-Lambda Rotor* in exactly the same wind field as for the reference turbine (blue lines in Fig. 16). Thus the maximum wind shear is the same, but the maximum wind speed at the top of the rotor disk is much larger for the *Hybrid-Lambda Rotor* due to its larger diameter. In fact, the wind speed at the top of the rotor plane increases from $12.5$ to $23 \, \mathrm{m \, s^{-1}}$ in 6 seconds and at the bottom, it decreases from $7.7$ to $0.8 \, \mathrm{m \, s^{-1}}$. Here, one should question whether

this is still a realistic shear event. Further, the wind speed at hub height is larger in general and marginally increases during the transient event, as the hub height of the *Hybrid-Lambda Rotor* is higher. All in all, the second approach can be considered as a site-specific wind field, as both turbines experience exactly the same wind. EWSV2 might be over-conservative, but we want to investigate the extremes in load case definitions.

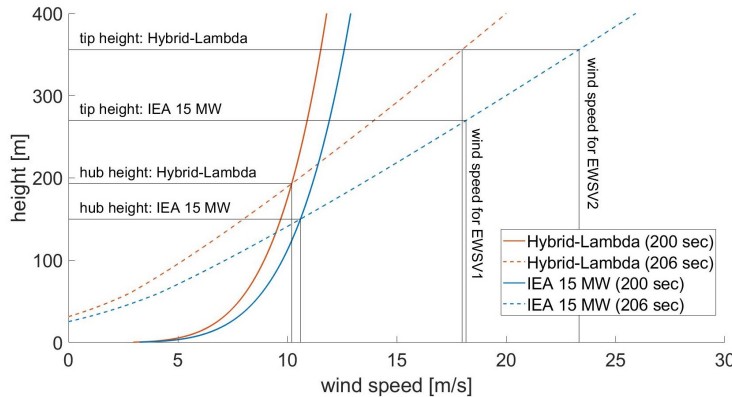

**Figure 16.** Extreme vertical wind shear profiles according to IEC 61400-1 (2019) for the *Hybrid-Lambda Rotor* and the IEA 15 MW reference turbine

  As seen in Fig. 15, the increase in flapwise RBM is very mild, with a factor of 1.09 for the turbine type-specific approach
(EWSV 1). As expected, the increase for the site-specific approach (EWSV 2) is larger, with a factor of 1.2. Still, it can be considered a great advantage that the increase in flapwise RBM is relatively low for blades of that size. Here, the low induction design for the outer section of the blade shows additional benefits. This is the part of the blade that experiences the highest wind speeds in an extreme shear event and it is beneficial if it is designed for a lower loading.
  To support this hypothesis, we want to have a deeper look at the out-of-plane forces distributed over the blade span in Fig. 17.
For this investigation, the site-specific wind field (EWSV 2) was used and the azimuth positions of the rotors were set in a way that both turbines have a blade pointing up at the time of the maximum wind shear (at 206 s). In Fig. 17, we display the force distribution before the transient shear event happens (at 200 s) and at the time of maximum shear (206 s). The non-dimensional lever arm ($L$) of the resulting bending force is calculated according to Eq. (8) and is indicated with the black vertical lines. Here, $f$ is the aerodynamic out-of-plane force per unit length, $x$ is the dimensional position along the blade span and $l_b$ is the
length of the blade.

$$L = \frac{\int f(x)x\,dx}{l_b \int f(x)\,dx} \tag{8}$$

The force distribution of the *Hybrid-Lambda Rotor* shows the characteristic decrease in the outer blade section with a maximum at approx. 65% blade length and a lower lever arm of the resulting bending force of approx. 55%. In comparison, the force distribution for the reference turbine increases more or less linear almost until the tip, which leads to a lever arm of approx.
65%. The maximum out-of-plane force per unit length is even 1.5 times larger than for the *Hybrid-Lambda Rotor*. Note that the

*Hybrid-Lambda Rotor* operates at a lower thrust coefficient at rated wind speed compared to the reference turbine. In addition, we investigated the transient wind shear event at $v_{shift}$ to ensure a similar thrust coefficient for the two turbines. But this load case leads to less severe ultimate loads and is not shown here in consequence.

Next, we are looking at the time-resolved change of the force distribution during the extreme shear event. For the reference turbine, the radial gradient of the force distribution increases with increasing wind shear. For the *Hybrid-Lambda Rotor*, the characteristic kink in the force distribution leads to lower maximum out-of-plane forces per unit length, even in the transient case. The non-dimensional lever arm is only slightly increased during the event and still much lower than for the reference turbine. We further found the increased flexibility of the *Hybrid-Lambda Rotor* beneficial in this case. The increasing wind shear

leads to a downwind deflection of the blade which reduces the relative inflow in the outer part of the blade and consequently reduces the load overshoot.

All in all, the lower non-dimensional lever arm and the dynamics of the blade contribute to a beneficial behaviour under extreme shear events, which leads to only mild increases of the flapwise RBM compared to the reference turbine.

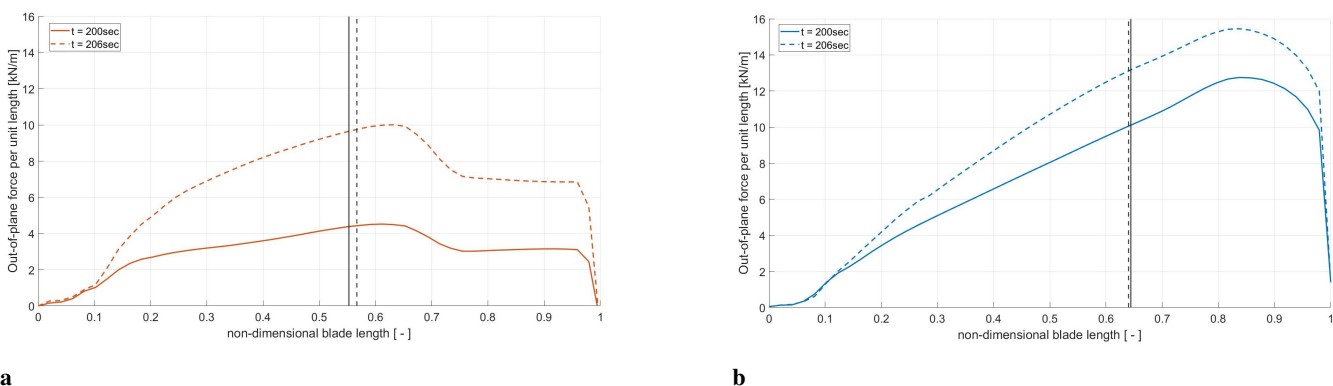

**Figure 17.** Out-of-plane forces per unit length under extreme vertical wind shear (EWSV2): For the *Hybrid-Lambda Rotor* (a) and for the IEA reference turbine (b). Resulting lever arms indicated with black vertical lines.

### 3.5   Techno-economic evaluation

After discussing the aerodynamic characteristics of the *Hybrid-Lambda Rotor* and setting up a structural model for the blades and the tower, we can combine those findings in a techno-economic evaluation. We calculate the gross AEP (neglecting availability, grid and wind farm losses) for three different offshore sites with the corresponding Weibull distributions shown in Fig. 18. First, we chose a reference site in the German North Sea at the north-west end of zone 4 (according to Federal Maritime and Hydrographic Agency of Germany (2020)) where the application of 15 MW offshore wind turbines is expected in the near

future. The Weibull distribution is derived by averaging wind data from the New European Wind Atlas for 10 years. A generic US East Cost site serves as a second reference site, as mentioned in the reference turbine report by Gaertner et al. (2020). A

third site is chosen to mimic the future wind conditions in the centre of the German bay when the increasing number of installed wind farms influence the wind resource in the boundary layer. Here, a Weibull distribution is adopted from Pettas et al. (2021) where data from the measurement platform Fino 1 is analyzed after the beginning operation of upstream wind farms.

First, we want to discuss how the modified power curve corresponds with the annual energy production. Figure 19 shows the gross energy yield per wind speed bin together with the Weibull distribution of the cluster-wake affected reference site. It becomes clear that for wind speeds less than $10 \text{ m s}^{-1}$, the energy yield is significantly increased by the *Hybrid-Lambda Rotor*. The financial losses due to derating the concept turbine for wind speeds greater than $15 \text{ m s}^{-1}$ are very mild due to the

low wind speed probability and the low market value of wind power.

As explained in Sect. 1, the AEP should no longer serve as merit functions because wind power will be more valuable on light wind days. To evaluate the results from the stationary BEM simulations, we calculate the economic revenue by weighting the energy production in each wind speed bin with the market value, as predicted by May et al. (2015) and visualized in Fig.

19. The results for the three reference sites are reported in Table 3 and the investigation points out a clear trend. The increase in economic revenue is always greater than the increase in AEP because the energy yield in light winds is more valuable on the energy system level. The benefit of the concept further increases if the future development of the offshore wind resource is considered. The power characteristic of the *Hybrid-Lambda Rotor* becomes even more attractive if the best sites are occupied and if cluster-wake effects need to be considered. In this case, AEP and economic revenue increase by 21% and 30% respectively,

when compared to the reference turbine. Enlarging the rotor diameter always leads to an increased AEP. Therefore we further compare the *Hybrid-Lambda Rotor* to the scaled version of the reference turbine with conventionally applied peak shaving, limiting the same maximal flapwise RBM (see dashed green line in Fig. 9). Considering the cluster-wake affected wind speed distribution, the AEP can be increased by 3% and the economic revenue by 4%.

Considering costs usually leads the designer to the objective function LCOE. But this metric neglects the variability of the market value of wind power and should not be used to assess turbine concepts that aim on providing a demand-oriented power feed-in. When evaluating the aero-structural optimization on an energy system level, we use the cost of valued energy (COVE). This metric is defined by dividing the annualized costs by the produced energy weighted with the normalized market value at the time of production as described by Simpson et al. (2020). With a cost model implemented in *WISDEM*, the annualized

costs are calculated. Further, the energy yield per wind speed bin is weighted with the expected normalized market price of wind power in the corresponding wind speed bin, as depicted in Fig. 19. Figure 20 shows that the COVE could be reduced by 13% with the initial blade and tower design (comp. with Fig. 11) and by further 3% with the structural optimization of the blade and the tower for the cluster-wake affected wind speed distribution. This figure also includes the LCOE to give an insight on how much of this reduction can be attributed to cost and AEP optimization versus the adaption of the market conditions.

A breakdown of the costs for the most important turbine components is shown in Fig. 21. Obviously, the largest increase in costs compared to the reference turbine is seen for the blades, since this is the part that increased the most in terms of size and

complexity. In fact, the costs of a blade increased by a factor of 2.8 (equals $n^{3.37}$). Related to the much heavier blades and the increased aerodynamic loading also the pitch system needs to be sized properly. Hence, the pitch system (plotted for all three blades) sees the second highest increase with a factor of 1.8, compared to the reference turbine. The tower costs increased by a factor of 1.2. The costs for the direct drive generator have the largest share of the total turbine costs and the derived generator costs for the reference turbine are comparable with the findings of Barter et al. (2023). For the *Hybrid-Lambda Rotor*, they increased by a factor of 1.22 since the rated generator torque increased. These numbers should only indicate an approximate trend of the cost breakdown, since the cost model in *WISDEM* relies on simplified scaling rules coupled to empirical datasets. For more insights, sophisticated models need to be set up for components like the pitch and yaw system or the generator.

**Table 3.** Weibull factors (A,k), AEP and annual economic revenue expressed as ratio to the reference turbine (ref.) for three reference sites

| | A | k | Ratio: Hybrid-Lambda/ref. | | Ratio: Hybrid-Lambda/scaled ref. (conv. peak shaving) | |
| --- | --- | --- | --- | --- | --- | --- |
| | [m s$^{-1}$] | [-] | revenue | AEP | revenue | AEP |
| German North Sea zone 4 | 11.48 | 2.22 | 1.21 | 1.11 | 1.03 | 1.02 |
| Generic US East Cost site | 9.77 | 2.12 | 1.27 | 1.17 | 1.036 | 1.027 |
| Fino 1 cluster-wake affected | 8.96 | 2.06 | 1.30 | 1.21 | 1.04 | 1.03 |

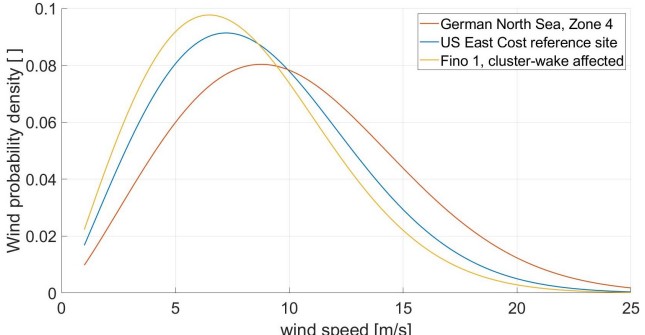

**Figure 18.** Weibull distributions for three reference sites

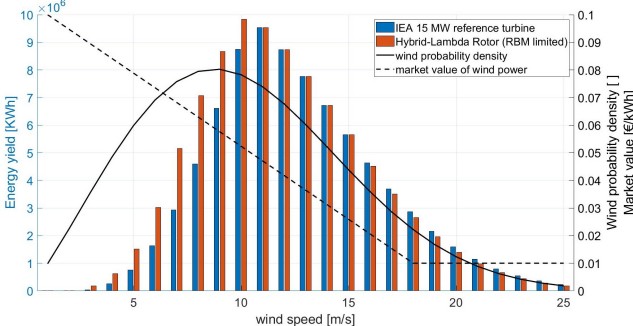

**Figure 19.** Gross energy yield, wind speed distribution (Fino 1 cluster-wake affected), market value of wind power

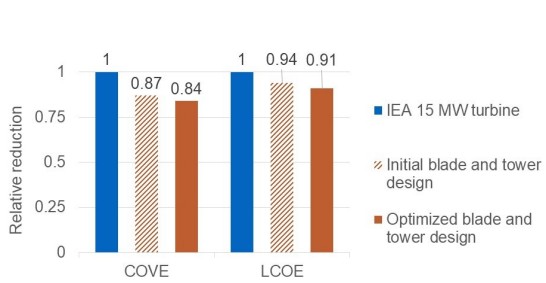

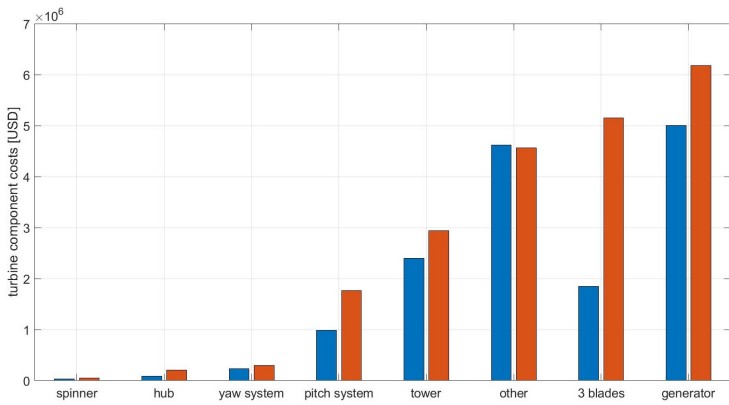

**Figure 20.** Reduction in cost of valued energy and LCoE relative to the reference turbine for the cluster-wake affected wind speed distribution

**Figure 21.** Estimation of turbine component costs for the IEA 15 MW (blue) and the optimized *Hybrid-Lambda turbine* (red)

## 4 Discussion

Designing blades for wind turbines usually follows several objectives and involves a multi-disciplinary approach. The novelty of our method is the dramatic reduction in the specific rating for offshore turbines. Moving away from designing the blade for one single optimized operational point, we include the application of peak shaving in the design process and introduce a design for two TSRs. This enables a unique change in the angle of attack distribution over the blade span when switching between the operating modes without complicating the technical implementation with additional actuators on the blade. Combined with an adapted controller, the design driving loads in transient DLCs can be reduced. The introduced design methodology should be seen as a basis for a wide range of blade design problems and can be customized accordingly. The exemplary solution worked out in this paper, a 15 MW turbine with a specific rating of $180 \text{ W m}^{-2}$, should only illustrate the possibilities of the design methodology. Further, we draw the comparison once again to existing offshore turbine designs that feature specific ratings from $350$ to $450 \text{W m}^{-2}$, as reported by Borrmann et al. (2018). Other design problems will lead to other solutions, including different specific ratings. Moreover, the choice of design TSRs can be adopted and the fact that we chose a step-wise implementation along the blade span with a sharp transition region should not hinder the designer's creativity. Alternatively, a continuous distribution of design TSRs along the blade span would be a possible extension of the methodology.

The *Hybrid-Lambda* concept can play an important role in the future path of the deployment of offshore wind energy. The increased power production in light winds will lead to a more steady power feed-in and will increase the value of wind power in a broader context of the future energy mix. This leads us to the question of the objective functions for the given design problem. Optimizing for the lowest LCoE on a single turbine level, as mostly done in the past, will lead to turbines with increased rated

power and medium to high specific ratings. As pointed out by many other studies (e.g. Dykes et al. (2020), Simpson et al. (2020)), there is the need for a change of mindset in the design objectives. The variability of the market value of wind power can no longer be neglected and objective functions like COVE, "system levelized cost of energy" or "value factor", just to name a few, as described in Simpson et al. (2020), should be included in future design problems. Further, wind farm related design objectives like wake-turbine interactions, wind farm control and large cluster-wakes will influence forthcoming optimization objectives and might drive the designs in the direction of lower specific ratings and lower induction factors.

Placing the *Hybrid-Lambda* concept in the state of the literature, we see common trends for several design variables, especially the axial induction factor and the specific rating. In the strong wind mode, the distribution of the axial induction over the blade span shows similarities with the one derived by Jamieson (2020). This is the analytically derived distribution that leads to the most possible extension of the rotor radius by incorporating a low induction design that features constant loading compared to a conventional design. In terms of the specific rating, the *Hybrid-Lambda* concept follows similar objectives as the *DTU Low-Wind-Turbine* for onshore siting by Madsen et al. (2020), but the innovations in the presented design methodology enable the application for offshore sites. A simple transfer of a low specific rating turbine with a conventional blade design to offshore sites would be inherently inefficient as the reduced aerodynamic efficiency in the upper partial load range would coincide with the maximum wind speed distribution and associated high losses in the energy yield. The idea of designing a blade for two TSRs was first mentioned by Wobben (2001). However, the main objective of this design was the reduction of stall effects in turbulent wind conditions.

One of the main pillars of the *Hybrid-Lambda* concept is the ability to change the angle of attack distribution over the blade span by switching between the operating modes. In fact, this change seems to be very little as we are tweaking the distribution by only a few degrees. Although one degree in the change of angle of attack can mean a lot to an aerodynamicist, these values are computed in steady-state simulations and neglect blade torsion as per definition of the structural methodologies used in this paper. Thus, we want to raise the question of whether controlling one degree in the angle of attack is at all feasible in a real application of a blade with 158 m length. Dealing with atmospheric turbulence, blade torsion, extreme shear events, low-level jets and other complex phenomena in offshore applications raises the question of whether those effects would simply dominate the uncertainties in the operation of such a blade. An answer that we owe in this paper. The concept could possibly be supported and the effects of the angle of attack change could be even enhanced by the usage of distributed aerodynamic control elements on the blade. Trailing edge flaps could increase the change in lift difference and leading edge slats could enlarge the operational range of the airfoils by delaying stall effects and allowing for higher angles of attack. Nevertheless, the authors decided to carry out this study without the usage of active control elements on the blade. Additional actuators introduce many problems, especially on very large blades, like additional masses, lightening issues, increased costs and reduced reliability. Although those add-ons could mitigate the dynamic loads when the turbine is in operation, still, storm events are critical design load cases. The presented design method leads to only 20% increases in flapwise RBM for the DLCs with extreme transient shear events. But storm events with a large yaw misalignment still lead to an increase of 40%, compared to the respective ultimate load from the

reference turbine. Nevertheless, the absolute value of the flapwise RBM is lower for the storm events than the maximum values for normal power production and extreme shear events. Limiting the loads in storm events is a difficult exercise as the turbine is not in operation and the usage of the control system is limited in the occurrence of faults. Nevertheless, we see a benefit in the slender blade design of the *Hybrid-Lambda Rotor* as the shorter chord length leads to reduced lift forces. Especially the outer part of the blade has a high lever arm and contributes the most to the RBM. But the slender design of the outer region helps to reduce the storm loads on the blade.

The consequences of the above-mentioned limitations of the methodology can be used as a guideline for further research. The blade design needs to be extended by including blade torsion, which also opens new opportunities for further improvements in the load-limiting techniques by using bend-twist coupling. Extending the structural methodology would allow us to calculate the blade torsional deformation under specific operating conditions. An iterative step back to adjust the design twist could optimize the angle of attack distribution for a given operational point, introducing an increased loading for lower wind speeds and a further relief for higher wind speeds. Finally, all those design features will not lead to a viable blade design as long as the aeroelastic stability is not given. As pointed out by Branlard et al. (2022), aeroelastic stability is a critical issue for very large wind turbine blades and it can easily overshadow other design drivers.

As above-mentioned, the analysis does not include fatigue loads. This is certainly a very important topic for large rotors. But in our opinion, including fatigue would only make sense if the conclusions lead to adjustments and improvements in the design. Achieving a fatigue-save design could further require adapted controller strategies (like individual pitch control, model predictive control or lidar-based feed-forward control) and a thorough implementation in the aero-servo-elastic simulations, which goes beyond the scope of this publication.

As a final point, we discuss the application of design load cases to rotors of such size. In Sect. 3.4.4, we introduced two wind fields for the event of extreme vertical wind shear. Given the implementation of the wind profile as equated in the IEC 61400-1 (2019), a larger turbine will be tested with lower shear. Across rotors with different radii, the maximum wind speed at the top of the rotor disc is kept constant rather than accounting for larger spatially distributed wind speed changes. On the other hand, the investigations above made clear that placing a large turbine like the *Hybrid-Lambda Rotor* in the same wind field as the reference turbine barely leads to a realistic load case, neither, considering a wind speed change at the top of the rotor disc from $12.5$ to $23 \mathrm{~m~s}^{-1}$ in 6 seconds and wind speeds close to zero at the bottom. Further metrological investigations will reveal what special inflow conditions very large wind turbines will need to face. Those are operating in regions of the atmospheric boundary layer, like the transition to the laminar Ekman layer, where the interactions with the rotor are not yet fully understood.

## 5 Conclusions

In this paper, we introduced an innovative design method for offshore wind turbines. It can be applied to any given design problem that aims for a tremendous reduction of the specific rating to enable an increased power feed-in for low wind speeds while maintaining the design driving loads. The design method integrates the application of peak shaving into the design process rather than accepting peak shaving as a necessary evil. Designing the blade for two different TSRs gives the opportunity to switch between a light and a strong wind mode during operation. In the latter operational mode, peak shaving is applied very efficiently, thanks to a beneficial change in the AoA distribution. This can be achieved with careful consideration of three aspects: The change in the inflow angle when lowering the TSR, a twist offset applied to the inner blade section and simultaneous pitching during the transition to the lower TSR. The power characteristics of the turbine better match the demand in the power system as the energy yield in light winds is significantly increased. Putting this into the context of the future development of the wind resource, the advantages become apparent, as wind probability distributions will shift to lower wind speeds due to the increasing clustering of large wind farms in several regions. Although the peak shaving region coincides with the maximum of the Weibull distribution for offshore sites, the losses are mild thanks to the efficient load reduction technique and the AEP is increased significantly. As energy will be (and already is) more valuable in light winds, the concept profits from a low COVE.

The presented design method was elaborated on a 15 MW offshore wind turbine to illustrate the concrete benefits. Comparing the *Hybrid-Lambda Rotor* to the reference turbine, the AEP and the economic revenue can be increased by 21% and 30%, respectively. With an optimized blade and tower design, the COVE could be reduced by 16%, compared to the reference turbine. We further compared the *Hybrid-Lambda Rotor* to a conventional blade design (upscaled to the same rotor radius) with a simple peak shaving application that limits the same maximum flapwise RBM. In this case, the *Hybrid-Lambda Rotor* can still outperform the conventional design by 3% in terms of AEP and by 4% in terms of economic revenue.

Using state-of-the-art design tools, we showed that it is possible to set up a consistent system-based turbine model for such an innovative rotor concept. The aero-structural optimization resulted in a blade that is 14% lighter compared to a geometrically scaled blade of the same size. Aeroelastic simulations showed the ability of the controller to change between the operating modes in turbulent wind and the *Hybrid-Lambda Rotor* was further compared to the reference turbine in a set of design load cases. This comparison shows promising results for the flapwise RBM, blade-tip-to-tower clearance and thrust. The flapwise RBM can be limited to the same maximum value as the reference turbine in power production DLCs. In extreme shear events, the flapwise RBMs are 20% higher and in storm events 40% higher compared to the reference turbine. Thrust and blade-tip-to-tower clearance could be limited to the same value as for the reference turbine. The increase in edgewise RBM and tower-top movements show open challenges in designing wind turbines of this size. It further becomes clear that advanced control strategies are needed to further limit the dynamic loads and to improve the switching between the operating modes. Nevertheless, even with a basic controller, the *Hybrid-Lambda Rotor* shows significant advantages in reducing the loads, although the swept rotor area is increased by 84% compared to the reference turbine.

*Data availability.* For the *Hybrid-Lambda Rotor*, we plan on providing the turbine model in the windIO format, the simulation model in OpenFAST and operational parameters in tabulated format in the near future.

*Author contributions.* DR realized and implemented the idea of the *Hybrid-Lambda Rotor*, added the conceptual idea of the change in the angle of attack and axial induction distribution, carried out the investigations and wrote the paper. FB assisted in developing the conceptual idea of the rotor concept. VP realized the initial controller for the OpenFAST model and supervised further controller developments. MK suggested the conceptual idea of designing a low specific rating rotor for two TSRs, contributed with fruitful discussions from an early planning stage and supervised the investigations. All co-authors thoroughly reviewed the paper.

*Competing interests.* There are no competing interests present.

*Acknowledgements.* The work presented in this paper was funded by the Deutsche Forschungsgemeinschaft (DFG, German Research Foundation) – Project-ID 434502799 – SFB 1463.

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
