# Peer review of "Hybrid-Lambda: A low specific rating rotor concept for offshore wind turbines"

_Wind Energy Science, 2023_

## Author Comment (AC1)

Dear Amy Robertson,

We have the pleasure of submitting our revised paper "Hybrid-Lambda: A low specific
rating rotor concept for offshore wind turbines" (wes-2023-72) for consideration in the
journal Wind Energy Science.

We are very grateful for the constructive feedback with lots of valuable suggestions
from the editorial team and the reviewers which helped to improve our paper. In
short, we want to highlight the major changes and additions:

- Design and optimization flow chart (Fig. 3.)
- Additional study on applying the *Hybrid-Lambda* control strategies to a
conventionally upscaled rotor (Fig. 9)
- Additional study on components costs and LCOE in addition to COVE (Fig. 20
and 21)
- Revision of Fig. 15, now showing absolute loads for all DLCs
- Restructuring and better explaining the concept of peak shaving, the transition
between the TSRs and the twist offset (Sect. 2.1 and 3.2)
- Added a plot to compare the stiffness and mass distribution along the blade
span with the reference turbine (Fig. 13)

Furthermore, we have made all the necessary requested changes, and have
addressed all comments of the reviewers (printed in black) in the detailed response
below.

Our responses to the referees are written in green.

Reformulated or added phrases for the revised manuscript are cited with blue fonts.

Line, figure and table numbers in our answers are according to the revised
manuscript. Line, figure and table numbers in the referees' comments are according
to the initial manuscript. All new and updated figures are appended to this authors'
response.

We feel that based on the reviewers comments our paper has been sharpened and
improved, especially in terms of clarity, readability and additional considerations, and
now meets the required standards to be published. If any responses are unclear, or if
you wish for additional changes, please let us know.

Sincerely,

Daniel Ribnitzky

- On behalf of all authors -

**Referee 1:**

*General comments*

The manuscript addresses the philosophy and methodology for rotor re-design to achieve a turbine that is better suited for electricity markets with high wind-energy penetration. Subsequently, the methodology is applied, and the resulting example design is evaluated on main performance indicators. The research is well motivated and introduced, with a clear description of the objectives. The main design philosophy is clearly argued and described. However, the methodology has a few complicated aspects that are challenging to understand. Particularly the aspect of pitching and how that influences the design of the inner blade section requires very much attention from the reader to grasp and only emerges gradually throughout the story. Likewise, which variables are optimised and how is not described in one place. In my opinion, the manuscript would benefit from restructuring this, for which I have some suggestions below, under 'Specific comments'. The results are interpreted fairly, with sufficient criticism, properly supporting the final conclusions.

On the principal criteria for WES publications, I would evaluate this manuscript with:

Scientific significance: Excellent

Scientific quality: Mostly excellent to good, and fair for the treatment of transition/pitch/optimisation

Presentation quality: Mostly good, for a challenging topic to explain, and again fair for the treatment of transition/pitch/ optimisation

Although a rather extensive section with specific comments follows, I would like to stress that I find the research very valuable and very well executed. I just want to share my ideas with the authors to stimulate them to see the work from a slightly different perspective. I'm happy with whichever way they use this information.

We thank the reviewer for the comprehensive, yet positive and encouraging feedback. We feel that the paper essentially improved by clearly marking the design variables and providing an overview of the design process with Fig. 3. We further improved the description of peak shaving and the twist offset and how this influences the blade design, as described below in more detail.

*Specific comments*

**Design philosophy and methodology (of aerodynamic design and control)**

I apologise up front for the lengthy discussion of this aspect. However, the authors know how many variables interact in the performance of a rotor, let alone in its design, and they have ample experience in trying to convey that to others. My struggle to provide clarity here will probably resemble theirs, so I hope this gives me some
leniency.

There are a few aspects of the descriptions of the design that I found difficult to follow.
For instance, several design choices are explained and motivated during the execution
of the design activities, while they are already touched upon earlier in the description
of the methodology. There turns out to be a strong relation between the final control
philosophy and the aerodynamic design for low-TSR / strong winds. However, this
control philosophy only becomes clear in section 3.2, while several references to its
consequences are already used in the descriptions and clarifications in chapter 2 (e.g.
lines 122-132, 152-160) and section 3.1 (e.g lines 221, 250-252).

Currently, the rotor design methodology starts with the principle of having three
regions: a light wind / high TSR region, a strong wind / low TSR region and a peak
shaving region (which is introduced on line 119, without explicitly describing how peak
shaving is done). At this point, the reader perceives these regions as being fully
separated. During the transition from light winds to strong winds the TSR and the
induction drop instantaneously, so the RBM drops instantaneously as well. Therefore,
pitching would only need to be applied at even stronger winds, when the (separate)
peak-shaving region starts. This would allow for a straightforward design of the inner
blade for zero pitch and at the optimal AoA, for low TSR. This is also how it is described
in figure 1, right (apart from the dual goal for the induction factor).

However, the final control philosophy introduces a longer transition, due to the choice
of keeping rotational speed constant in a transition region (rather than reducing it
instantaneously). Consequently, the blade needs to be pitched in the transition region,
and the pitch angle in the strong-wind region is no longer zero. Inherently, the
transition region is extended by this up to wind speeds where peak shaving is needed.
Therefore, there is no 'clean' strong-wind / low-TSR region, but this region is
immediately combined with peak shaving. It also seems that the term peak shaving is
often used loosely, to imply both control regions 2.2 and 2.3. As seen in figure 6, in
the strong-wind / low-TSR region the blade is not at constant pitch, so there are no
'unique' design conditions for the aerodynamic design for this region (i.e. with
constant TSR and constant pitch). These differences with the primary philosophy
explained in relation to figure 1 where initially very confusing to me.

We added a paragraph in Sec 2.1 to clearly address that switching between the
operating modes is realized by a continuous reduction in TSR. We further clarified that
a reduction in TSR alone is not enough to limit the loads and properly defined the
term peak-shaving:

The transition between the operating modes introduces a new control region since
the switching of the TSR is not a sudden change rather than a continuous reduction
in TSR. In this paper, it is realized with a constant rotational speed (rpm) in region 2.2
as shown in Fig. 2. The reduction in TSR alone (with a constant rpm) is not enough to
limit the loads. On the contrary, it is part of the design methodology to combine a reduction in TSR and pitching to feather for load limitation as further analysed in Sec.
3.1. Consequently, the so-called strong wind mode cannot be described with a
constant pitch angle. With increasing wind speed the pitch angle is gradually increased
towards feather to limit the flapwise RBM. This action will be referred to as peak-
shaving in the following.

It stands to reason that an extended transition region is beneficial. Without it, the drop
in TSR will be accompanied with a drop in BRM, but also a drop in power. Most likely,
power can be maximised in a transition region, if the BRM remains at its constraint.
This can be achieved with 1. constant speed and pitching (as chosen), 2. constant
(zero) pitch and a gradual reduction in rotational speed, or 3. a combination of speed
and pitch changes. Choosing for one of the first two (simpler) options is reasonable.
Unfortunately, this kind of logic using the level of system parameters is not provided.
Instead, the more complicated, implicit evaluation of the effect of speed and pitch on
load distribution is given (later), leaving it up to the reader to judge if this achieves the
desired global behaviour.

We thank the referee for these additional ideas on how to perform the transition
between the operating modes. We added a paragraph about these options in Sec. 2.1.
In fact, we ran optimization routines (in steady and uniform inflow and with rigid
structures) to find the best combination of TSR and pitch in the transition region (and
up to rated wind speed), that constrain the maximum flapwise RBM and maximize the
power output. We did not address this solution in the paper since the advantages in
terms of power output were only marginal.

Note, that the transition of TSRs could also be realized in different ways (e.g. reduction
or gentle increase in rpm). In fact, the optimal combination of TSR and pitch for the
transition region can be found by constraining the flapwise RBM and searching for the
optimum in the power coefficient. These optimization routines resulted in a gently
increasing rpm until rated wind speed. However, for all wind speed bins, the increase
in the power output was never larger than 0.5 % of rated power compared to the
constant rpm solution presented here. Consequently, the aforementioned alternative
for the transition region is not presented in this paper.

For better understanding of this approach, I recommend moving at least the top-left
of figure 6, to section 2.1. This speed control is so straight-lined, that it seems to be
more like a pre-meditated aspect of the design methodology than a consequence of
the execution of a design iteration. This graph will help understanding of many
aspects of section 2.1 that are currently unclear. Understandably, the authors did
learn from their early design experiments for the tuning of this graph (such as the
onset of the speed reduction at 15 m/s), but the same applies to the a-priory choice
of TSR 9 and 11. I think it is also necessary to already explain the consequence of this
speed control for the extent of the transition region, for pitch control, for the non-zero
pitch of the low-TSR design and for the non-constant pitch in strong winds (during
low-TSR operation in region 2.3). A schematised version of the bottom-left of figure 6
could be used for that as a qualitative pre-analysis. The shapes of the curve can easily be described with qualitative arguments for all regions. As a follow up of this
description, it can then be clarified that the inner blade section is designed for a
different/non-zero pitch angle and how that pitch angle will be determined during the
design process. It would help to add this change in design pitch-angle to figure 1.

We agree that the rotational speed schedule is important for understanding the
following design steps and moved Fig. 2 to Sec. 2.1, as suggested.

Although the previous description is reverse engineered from the manuscript, I'm
fairly sure this captures (part of) the rationale of the authors. The concurrent change
of TSR (for constant-speed operation) and pitch angle, will therefore naturally lead to
the effects described in figures 4 and 5. As such, those graphs could support the
choice for constant speed plus pitch increase, instead of constant pitch with speed
reduction. However, the bottom-up approach that starts with the graphs in figures 4
and 5 and ends with exactly constant speed operation is neither convincing nor clear.
If the authors agree with (some of) this analysis, I suggest that they restructure the
story along similar lines of reasoning.

We agree that cause, reason and prerequisites were not always clearly separated and
indicated in the initial manuscript. We therefore moved the description of the
rotational speed schedule up front to Sec. 2.1, as suggested.  The additional changes
to the manuscript are printed in the two answers above.

Up to this point, I agree with the overall philosophy for design and operation. On top
of this, the authors introduce two aspects, which I'd like them to either reconsider, or
support more clearly. These aspects are the twist offset towards stall for the inner
blade and the dual goal for its induction factor (0.33 in high-TSR operation and 0.21 in
low-TSR operation). I will start with the dual goal for induction, as this is easier to
address. The principle of the design is to provide power by the outer blade section in
light winds, to reduce loads on this section in strong winds and to let the inner section
take over power production in strong winds. Power production in strong winds is
considered to be important for offshore wind turbines, since these have a high
probability of occurrence. Hence the interest of the authors in good peak-shaving
performance. All these intentions, given by the authors, are counteracted by
prioritising the induction factor optimisation of the inner blade for power production
in light winds. As above, I would agree if the authors used an analysis of what happens
to the induction factor as argument for the choice of a constant speed-increasing pitch
transition: if the inner blade section is designed for an induction factor of 0.21 with a
positive pitch angle, then it will have a higher induction factor at zero pitch and high
TSR, which is a welcome advantage. This advantage is again a natural consequence of
the pitch and TSR actions. The need to fine-tune this with a dedicated design for an
induction factor of 0.33 in off-design operation is insufficiently argued.

The referee is correct in her/his description of the dual goal for the induction factor of
the inner blade section. In fact, the blade design for the inner section is driven by two
objectives. First, a traditional axial induction factor distribution of constant 0.33, as the aerodynamic optimum for power production in light winds (high TSR, fine pitch of
-0.8°). And second, a low induction rotor design of around 0.21 with decreasing axial
induction factor towards the tip in strong winds (low TSR, positive pitch) for load
reduction. For power maximization in light winds both parts (inner and outer blade
section) are important. While in strong winds first of all the inner part is important,
since the outer part does not contribute much anymore to the power production and
the loads. This somehow delicate design compromise is definitely not achievable with
a conventional rotor design and the fact that we could achieve it already explains how
we "integrate the application of peak shaving into the design process" which was also
questioned in other comments by the referees. We added a justification to the
description of the light wind mode in Sec. 2.1:

The inner part of the rotor operates like a conventional rotor with an axial induction
factor close to 0.33. This is chosen in order to maximize the power output in light
winds. But the reader should bear in mind that this part is not operating at its design
point, as it is designed for a lower TSR of 9.

Then, the twist offset. If I'm correct, this offset is relative to its optimal AoA in low-TSR
operation, although that conflicts with the information in figure 1. Also here, in terms
of design philosophy this doesn't make sense in a first-order rationale: The primary
goal for the inner blade is power production in strong winds, so a compromise of the
design for low-TSR operation should be very strongly supported.

Here we at least partly disagree with the referee. Yes, the twist offset is applied relative
to the optimal AoA in the low-TSR operation. The twist offset defines the difference in
the axial induction factor between the light and strong wind mode for the inner blade
section, as well as the resulting pitch angles in the strong wind mode (e.g. how much
we need to pitch to limit the loads). In fact, the pitch angle of 2.2° at $v_{shift,end}$ almost
perfectly counterbalances the twist offset of -2.5°. Hence, the inner part of the blade
operates in it's optimal AoA (at this wind speed) and we don't see a conflict with Fig. 1.
We further have a different view on the described "primary goal for the inner blade".
The design is highly driven by compromise findings and the goals can't always be
categorized into primary and secondary. We would rather like to highlight the equality
of the two objectives: Power maximization in light winds by a drastic increase in rotor
diameter and limitation of the loads in strong winds with reduced power losses.

Furthermore, I'd like to go into the description of the effects of the twist offset, that
are used to argue its need. In much of the operational region of the aerofoils, the lift
coefficient depends nearly linearly on the AoA. Thus, pitching is almost equally
effective with and without the twist offset. Likewise, the effect of changing TSR on the
change in lift coefficient over the blade span (relating to figure 4) is hardly dependent
on a twist offset, since it is an effect on inflow angle: the change in AoA is not affected
by an offset. Furthermore, for the system-level phenomena that are discussed, the
optimality of the AoA hardly matters, so the offset of the twist is effectively relative to
an arbitrary AoA. This also makes the discussion in lines 294-299 confusing or even
misleading: Optimum AoA only tells something about the lift over drag ratio. For this special design and for the many off-design conditions (dual TSRs, transition pitching
and peak-shaving pitching) it doesn't tell anything about the bigger picture for
induction, power coefficient or thrust coefficient. Because of the operation at high TSR
(with zero pitch) and at low TSR (with positive pitch), both the inner blade section and
outer blade section have fundamentally two operating AoAs. I would be more
concerned about how these two points are situated in the region between maximum
lift coefficient/stall and minimum/negative lift. If the margins to those are good, then
I would prioritise the optimum AoA under the principal design conditions (inner: low
TSR – outer: high TSR) and not the off-design conditions (inner: high TSR).

The twist offset indeed introduces very complex effects on the blade design, the
aerodynamics and the resulting control strategy. We apologize if this was not clearly
discussed in the paper and we tried to improve the comprehensibility in the revised
manuscript. But, we also want to clarify that the optimality of the AoA (lift to drag ratio)
indeed does matter. As described in Burton et al. (2011) (Fig. 3.26 in chapter 3.7.5.)
the lift to drag ratio has a non-neglectable influence on the power coefficient. The
referee is correct in stating that pitching changes the lift coefficient equally,
independent of the "starting-AoA", since the behaviour is linear throughout the
operating range. But, it is not equally effective (e.g. how much do we lose in terms of
power coefficient) since there is a difference if you pitch away from the optimum lift
to drag ratio or towards the latter. By introducing the twist offset it is possible to pitch
the blade to feather (for load reduction) while increasing the lift to drag ratio for the
inner blade section (keeping the aerodynamic performance high). With that we
realized exactly what the referee suggested: *"prioritise the optimum AoA under the*
*principal design conditions (inner: low TSR – outer: high TSR)".* This can be seen in Fig. 8
for the angle of attack in strong wind mode. Here, also the stall angle and the angle
for the maximal lift to drag ratio is indicated. We agree, that part of the mentioned
benefits (e.g. the change in the AoA distribution) could also be realized without the
twist offset, just using the reduction in TSR and pitching to feather. At the end, the
twist offset is also a tool to tune the difference in the axial induction factor between
the light and strong wind mode for the inner blade section and allows to use smaller
chord lengths. This further leads to a more slender, lighter and possibly cheaper
blade. We reformulated the last paragraph in Sec. 2.1 to address the advantages of
the twist offset more clearly:

Furthermore, we design the blade in a way that peak shaving is applied more
efficiently. The inner section is designed with a twist offset towards stall. This comes
with several advantages. The inner section does not operate in the design point in the
low wind regime. As it is twisted towards stall and operated at a higher TSR than it was
designed for, a fairly conventional induction factor of 0.33 can be reached, which leads
to an increase in the power coefficient in the low wind regime. The angle for the twist
offset is derived iteratively in stationary blade element momentum (BEM) simulations
to reach the desired axial induction factor of 0.33 in the inner section at the high TSR.
Using the twist instead of the chord length as a tool for this increase in the axial
induction factor allows to use smaller chord lengths which leads to more slender, lighter and possibly cheaper blades. Hence, the twist offset defines the difference of the axial induction factor between the light and strong wind mode for the inner part of the blade and it further influences the pitch angle at $v_{shift,end}$ that is needed to limit the loads. In fact, the pitch angle of 2.2° at $v_{shift,end}$ almost perfectly counterbalances the twist offset of -2.5°. Hence, the inner part of the blade operates in it's optimal lift to drag ratio at this wind speed, although the entire blade is already pitched to feather for load reduction. When peak shaving is applied, pitching shifts the inner section to operate at its aerodynamic optimum rather than moving away from it. It reaches its design point (an induction factor of 0.21 at the low TSR), which is beneficial for load reduction. In contrast, the outer section is now operated in a "pitched-to-feather-condition" and is greatly relieved. The limits to this methodology are negative lift and the stall angle. The latter is also plotted in Fig. 8.

I concur that the authors might prove to be correct in their arguments for deviation from the first design principles, to fine-tune the performance. However, there is so much going on, that I don't think it helps understanding the fundamentals. Obviously, the design principles of the Hybrid-lambda rotor can be combined with other philosophies, such as induction reduction towards the tip. However, a separation of effect would be beneficial for obtaining better insights. Induction reduction is here primarily achieved by the Hybrid-lambda design, and secondarily by the inner section design adaptations. Possibly, the authors already have experimented with the straightforward design approach and have found it to lead to unacceptable behaviour. In that case, it would be helpful to describe that more explicitly.

Thank you for the suggestion. Indeed, we tried to separate the effects as best as possible to identify the potential of each design decision (e.g.: Using a step-wise distribution with only two TSRs, rather than a continuous change in TSR; Separating aerodynamic and elastic effects at first; Separating instationary effects and the influence of controller tuning; etc.). However, we did not want to further complicate the reading by describing several design versions of the Hybrid-Lambda rotor (there were definitely many versions developed on the long path to the final design presented here). To provide a clear reference we introduced the scaled conventional blade which uses simple peak-shaving (only pitching to feather to limit the loads), as printed in line 369.

**Optimisation procedure**

The design methodology (chapter 2) describes how the blade is designed for a particular rotor diameter and doesn't describe if and how rotor diameter is optimised. Rotor diameter also doesn't appear as design variable in the optimisation methodology (of section 2.3), where these variables are declared on lines 175-176.

Section 2.3 describes the methodology for the structural design and optimization and the aeroelastic investigations. The rotor radius is not mentioned here since this is analysed in the previous aerodynamic design step. The design flow chart (added to

Sect. 2.1 of the revised manuscript and additionally displayed in the appendix of this authors' response) will help to clarify our workflow. We added to the beginning of Sec. 2.3:

To further investigate the feasibility of the Hybrid-Lambda Rotor we develop a structural model for the blade. The workflow described in this section is carried out after freezing the design output-variables rotor radius and the chord and twist distribution. A link back to the aerodynamic optimization was only performed for a few major design versions, as indicated in Fig. 3.

Perhaps what is described there is a nested optimisation (inner level), but that is not described. As it is, the value of 326 m for rotor diameter on line 216 comes out of the blue. Similarly, it isn't clarified in the methodology how the spanwise transition point will be determined. The effect of both variables is discussed later (lines 254-280), which implies that they are also design variables (according to line 213). It would be helpful to know in advance how these design variables are incorporated in the methodology.

We want to give the reader an impression of the size of the rotor up-front. We tried to keep the description of the methodology (chapter 2) as general as possible in order to provide a design idea that can be adopted to different wind turbine design problems (see lines 100-104). Starting with the results in chapter 3, we are explicitly explaining the concept on the basis of the worked-out example (15 MW, 326 m diameter). The effect of the rotor radius (and specific rating) is described later in lines 279-294. The effect of the spanwise transition point is discussed in lines 296-304. We believe it is meaningful to provide the reader with a short overview first (mentioning radius and spanwise transition point) and then explaining the effects later in the same section (3.1). We added a paragraph to the beginning of chapter 3 in order to clarify this:

In this chapter, we focus on the given use-case of the 15 MW offshore wind turbine, no longer generalizing the concept, in order to simplify the understanding. This means only one specific turbine diameter is presented here, although the influence of the rotor radius as a design variable was investigated and is further described below. We first address the resulting blade design and the influence of certain design variables. Table 2 summarizes general turbine parameters. The second part deals with loads, axial induction, angle of attack and power generation under steady and uniform inflow conditions. This is followed by the results of the structural design and the aero-servo-elastic investigations.

Along similar lines, lines 78-79 describe that the objective function (implied: for rotor optimisation) is COVE. However, the optimisation of the tower is only described later. It is not clear whether this tower optimisation is included in a global exploration or nested optimisation in this optimisation of COVE.

We clarified this in the design flow chart. The objective function for the blade
structural optimization is COVE, as stated in line 213. The objective function for the
tower design is the combined structural mass of tower and monopile, which is
mentioned in line 228.

On line 178, a stall margin is introduced as a constraint for the optimisation. It is not
clear how this is implemented, since the aerodynamic design methodology doesn't
(explicitly) accommodate that.

We further explained the stall margin constraint:

Constraints for the optimization process are tip deflection, blade eigenfrequencies
(must be above the rated blade passing frequency, 3P), the strains in the spar caps
and a stall margin. The latter would only be active if the change in the airfoil position
leads to an operating angle of attack larger than the stall angle of the respective airfoil
(chord and twist are not optimized in this structural design step).

Line 180 states that this optimisation is done for a wind speed of 6.9 m/s, but it is not
clear to the reader how this can be known. The wind speed at which the light-wind
mode ends even seems to be a consequence of the optimisation itself, considering its
dependence on rotor diameter.

Although the rotor diameter is not a free design variable in the structural optimization,
the wind speed $v_{shift,start}$ is explicitly calculated for every design iteration in WISDEM
(with the code changes applied by the authors). We added this description to the
methodology:

For each iteration the schedule of rpm, pitch, power, thrust and flapwise RBM over
wind speed is re-calculated. The considered load case for the constraints is a steady
inflow at the strongest wind speed in the light wind mode $v_{shift,start}$, as calculated for
each design iteration (in this case $v = 6.9 \frac{m}{s}$, $TSR = 11$, $\beta = -0.8°$).

All in all, I was somewhat confused about which aspects were optimised in a numerical
optimisation, which aspects were determined in an analytic design approach and
which aspects were designed with the authors in the loop. Relating to that, it wasn't
always clear in which order the various design variables were fixed. It would be helpful
to clarify that in the beginning, perhaps with a flow chart of the entire process. In
addition, it would help to categorise the variables in table 2 (fixed/chosen, design
variables, properties, ...). In the results, I propose to start with the discussion of rotor
diameter and spanwise transition (lines 254-280), since these are two high-level
system parameters.

We thank the referee for the idea of a design flow chart. We added this to the revised
manuscript for further clarification (see also the appendix of this authors' response).
We further added a description of the design flow chart to line 153:

The overall design and optimization workflow is illustrated in Fig. 3. The process can be explained in four steps: An aerodynamic blade optimization, an aero-structural optimization of the blade, a structural optimization of the tower and the aero-servo-elastic simulations. In the first step (aerodynamic optimization) the design variables are the transition point between the inner and outer blade section, the design TSRs, the design axial induction factors, the twist offset and the design angle of attacks. Once a reasonable design is established the influence of the rotor radius is investigated. In the second step (the aero-structural optimization) the design variables are the airfoil positions and the spar cap thickness. When this step is converged the aerodynamic optimization is re-calculated once with the new airfoil positions. As a third step the tower and monopile are optimized for a fixed rotor design. The resulting turbine design is then investigated in aero-servo-elastic simulations.

We further classified the variables in table 2 to highlight the differences between optimized design parameters and predefined parameters. When explaining the effect of certain design variables in lines 278-317, we changed the order as proposed.

**Results**

The design is assessed on AEP, revenue and COVE. Although the design is intended to advance from LCOE optimisation, it would be interesting to add how well the new design and reference perform on that metric. This would help understand to which extent the new design is a conventional improvement on LCOE, and which part can be attributed to the adaptation to the market conditions. This is similar to the comparison between AEP and revenue, which is already made. In addition, it might be useful to show and discuss some cost results separately, and not only hidden inside COVE.

We included the LCOE in Fig. 20 and added:

This figure also includes the LCOE to give an insight on how much of this reduction can be attributed to cost and AEP optimization versus the adaption of the market conditions.

We further corrected a typo in the legend in Fig. 20:

 Initial blade and tower design

**Discussion**

There are good messages in the discussion. I would recommend discussing only aspects that are closely related to the proposed concept and the results of this study. Adding other concepts/technologies (such as actuators and bend-twist coupling) is not specific to this concept (or at least it isn't argued why a combination would be of more interest than for conventional designs). There are numerous other concepts that could otherwise be named as well.

We used the description of additional actuators and the accompanying disadvantages
to highlight the benefits of the Hybrid-Lambda rotor. We further like to mention the
bend-twist coupling since we believe that including blade torsion to the simulation
model only makes sense with a substantial redesign of the blade twist, accounting for
and counterbalancing the blade torsion. We therefore decided to keep theses
descriptions in lines 752-757 and 767-771.

In my opinion, the generalisation of the method to continuous variable-TSR operation
(with variable spanwise induction optimisation) is the most interesting part of the
discussion. It could be considered to dive a little deeper into this.

Our idea was to distribute the design TSR over the blade length with a continuous
function. This could enable three advantages. First, the steep gradient in the blade
design (twist and chord) would be reduced which simplifies the structural design.
Second, the axial induction and angle of attack distribution would be smoother. Steep
gradients might lead to additional trailing vortices and a continuous distribution might
be beneficial. Third, since lowering the operational TSR is a continuous control action
there would always be some part of the blade operating in its design point. The further
the operational TSR is reduced, the further this part would move along the blade
towards the root. Like this, one could generalize the concept to a continuous TSR
reduction towards rated wind speed, not using distinct light and strong wind modes
anymore. Since all these thoughts and ideas are speculations so far and we couldn't
find the time to implement such a design idea, we decided to not further explain the
idea in the paper. We will keep it in mind for further publications.

**Conclusions**
On line 710-713 you state that peak shaving is integrated into the design process. As
you have seen in my earlier comments, I found this part somewhat confusing. I
struggled with the use of the term peak shaving for both the transition region and for
the conventional peak-shaving region. Furthermore, the bottom-up argumentation
for the chosen control was difficult to follow. It didn't give a reproducible procedure
to merit the name 'integration in the design process'. To claim this integration, I would
like to see at least a stricter process for this particular part of the design approach,
such as could be given with a flow chart, a formal optimisation problem description
or graphs with dependencies on relevant design variables. As outlined above, in my
opinion you provide arguments for a sensible choice of operation in the transition
region, but that wouldn't go as far as a design process. As it is, you only show one
design point, with only circumstantial evidence that it provides superior performance
thanks to the claimed mechanisms. Perhaps a similar combination of speed and pitch
control can achieve similar performance for peak shaving with a conventional rotor
design.

Having said that, the conclusions provide a concise overview of the relevant insights
that have been achieved with this research.

It is one key aspect of the proposed design methodology to consider the fact that the blade will be pitched to feather before rated power is reached to limit the loads. Since the blade designer knows this already, it should be integrated in the blade design process beforehand. We agree that we didn't provide a clear evidence in the paper, about how this is done. We further thank the referee for the idea of comparing the performance with a conventional blade where both, pitch and TSR, are optimized when peak shaving is applied. We carried out an additional study and applied similar control optimization strategies to the scaled version of the IEA 15 MW turbine. This breaks up to what extent the improvements result from the change in the control strategy and to what extent from the integration in the blade design. We added the results to Fig. 9 and added a descriptive paragraph:

The green dashed line indicates the power curve of the reference blade that is geometrically scaled by the same factor and conventional peak shaving is applied to limit the flapwise RBM. This means only the pitch angle is set to a higher value to constrain the flapwise RBM while the rpm follows the design TSR. In contrast, the black dotted line represents the same blade (geometrically scaled IEA 15 MW) but peak shaving is applied in a similar manner as for the Hybrid-Lambda Rotor. This means for $v > v_{shift,start}$ the rpm is kept constant until the operational TSR is reduced from 9 to 7. For $v > v_{shift,end}$ the rpm schedule follows the TSR of 7 which is an arbitrary choice in this case and should be optimized in a detailed design study. In addition, the pitch angle is set for $v > v_{shift,start}$ in order to limit the flapwise RBM. In short, we are applying the Hybrid-Lambda control strategy to a conventional blade design. The results show that the power output can be greatly increased if the TSR is lowered in region 2.2 and 2.3 (compare green dashed and black dotted line in Fig. 9). Thus, peak shaving should not only be accomplished by increasing the pitch angle, but also by optimizing the operational TSR with respect to the load constraint (as also indicated by Madsen et al. (2020)). Since the results show that a reduction of the operational TSR is beneficial in the peak shaving region it makes sense to account for this fact already in the blade design which is integrated in the Hybrid-Lambda design methodology. Indeed, the Hybrid-Lambda Rotor enables even lower power losses in the peak shaving region since the TSR reduction is already accounted for in the blade design (compare solid red and dotted black line in Fig. 9). The turbine concept reaches its rated power at 10.2 $m\,s^{-1}$, which is 0.4 $m\,s^{-1}$ lower than the reference turbine.

**Smaller comments about the content (in order of appearance)**

- On line 53 a similar design philosophy from Wobben is mentioned. This is later discussed on line 656, where it becomes clearer in which sense that philosophy differs. It could be useful to clarify this already in the introduction.

We moved the description to the introduction, as proposed:

This concept follows the objective of reducing unintended stall effects on the blade of
a variable-speed turbine in gusty winds. It was not used to enable large rotors with
low specific ratings, as pointed out with the *Hybrid-Lambda* concept.

- Line 64 (and many other places): The authors use 'zero pitch' for the operation
of the blade at design conditions. This is implicitly defined on line 64. However,
many blade designers and control designers define the structural twist with
respect to zero twist at the tip and then use something like 'fine pitch' to get
the design twist at the tip. Thus, this offsets the definition of pitch from the one
used in this manuscript. It seems that even the authors confused themselves
about this, since figure 6, bottom-left, shows a negative pitch angle for high-
TSR operation. The chosen definition could be made more explicit (and used
consistently).

The term zero pitch should indicate only that the pitch angle is at zero degrees. In
many blade design studies the so called fine pitch, that leads to the maximum power
coefficient at design TSR, can deviate slightly from zero degrees, as it does in our
study. We clarified this, using the term "zero pitch" only if zero degrees are meant and
using "fine pitch" when the pitch for optimal $c_p$ is meant. We added to the description
of Fig. 2:

From 4 $m\ s^{-1}$ on, the rotor operates at the high TSR of 11 in the light wind mode and
a fine pitch angle of -0.8° which leads to the maximum power coefficient. This pitch
angle is called fine pitch since the pitch angle for optimal $c_p$ was derived after the
blade design was concluded.

- Line 257-258: The sentence 'If ... reached' is not so clear.

We added the respective control region numbers for clarification:

If the rotor radius is enlarged, the power output is increased before the limiting loads
are reached (e.g., in region 1 and 2.1). But at higher wind speeds, when peak shaving
is applied (in region 2.2 and 2.3), the blade must be pitched further and power losses
are more pronounced.

- The authors claim on lines 319-320 that the reduced thrust coefficient leads to
much lower wake losses. This cannot be known, since the effect of increased
rotor diameter cannot be ignored. The increase in rotor diameter will extend
the wake over longer distances and over a wider area. The next sentence
implies that actually more momentum is taken from the wind.

We addressed the wake losses of the *Hybrid-Lambda Rotor* in a separate publication,
which is accepted but not published yet (Ribnitzky, Bortolotti, Branlard, Kühn: *Rotor*
*and wake aerodynamic analysis of the Hybrid-Lambda concept - an offshore low-specific-*
*rating rotor concept,* JoP conference series, 2023). Results show an increased power
output on a two-turbine set-up even though the rotor radius is enlarged and even in a scenario of constant absolute spacing (compared to the IEA 15 MW). We added the
citation:

The wake losses of the Hybrid-Lambda rotor are addressed by Ribnitzky et al. (2023).
Results show significant advantages even in a scenario with constant absolute spacing
(compared to the IEA 15 MW reference turbine).

• Lines 353-362: Does Wisdem take the special care that is meant here? For
instance, this region would experience stress concentration. Is that accounted
for? Otherwise, the reduction in spar-cap thickness could be more related to
model simplification than to optimisation.

*PreComp* estimates equivalent sectional inertia and stiffness properties for 2D cross
sections with the help of a modified classic laminate theory. It's not a 3D finite element
model, hence gradients in the stiffness distribution in blade spanwise direction are
not considered. However, for each cross section realistic stiffness properties are
derived and the resulting material stresses are calculated. We further added the
stiffness distribution and compared it to the IEA 15 MW (as requested by the second
referee). We added a note to the manuscript:

Here, the reader should bear in mind that the structural solver *PreComp* is a 2D cross
sectional solver and does not account for stress concentration due to rapid changes
in the geometry in span wise direction.

• Lines 388-390: This description is ambiguous. In region 2.3 the blade has
variable pitch, so there is no unique c_P for this region. Could this be clarified?

This is correct, $c_p$ is changing with the wind speed in region 2.3. But, from steady state
simulations the desired $c_p$ is known as a function of wind speed (or in the given case
of eq. 6, as a function of rotational speed). We added the dependency in the equation
and added:

$$M_g = \frac{\pi R^5 \rho c_p(\omega)}{2\lambda^3} \omega^2$$

Note, that there is no constant $c_p$ in region 2.3 since the pitch angle is a function of
wind speed. Hence, the desired $c_p$ from steady state simulations is implemented as a
function of rotational speed.

• Lines 400-402: It is described that a conventional look-up table was not found
to perform sufficiently well. Could it be clarified whether this means that
something else has been implemented? This seems to be the implication, since
this section is about the controller design, and not about its evaluation.
Therefore, this doesn't seem to be simply an observation of performance, but
a reason for change.

We implemented two versions of the controller. The advanced controller with the load
feedback was only applied for the investigations in Sec. 3.4.2. We clarified this by
describing and naming both controller versions:

For the pitch controller two versions are implemented. The first version is referred to
as simplified controller and implements the transition of the TSR and a look up table
for the pitch signal for regions 2.2 and 2.3. This simplified controller is used for the
load case calculations in Sec 3.4.3. A second version is developed that features a
feedback from the flapwise RBMs, further referred to as load feedback controller and
it is applied in Sec. 3.4.2.

• Lines 403-404: Could the authors explain what is meant by 'minimal' and
'reduce' compared to what? The previous descriptions of prescribed pitch do
not seem to relate to the region where RBM load control is needed, or is it
(dynamically)? The later text (lines 412-413) implies that 'minimal' refers to the
steady-state pitch angle that was previously discussed. It would be helpful to
get this information first. Having said that, lines 457-459 state that this
controller is not used. Therefore, I would recommend removing this entire
description of the (dynamic) load controller.

As discussed above, the load feedback controller is used in Sec. 3.4.2 and should
therefore be described in the manuscript. Although there is a mismatch with the line
numbers the authors are guessing that the term "minimal pitch" is causing confusion.
We therefore changed the naming to reference pitch, which is meant to be the output
signal of the controller. We further added the controller region numbers where the
load limiter is usually active.

Thirdly, in parallel to these two functionalities, we implemented a load limiter (for
region 2.2 and 2.3). (...) As long as the RBM feedback is larger than the constraint, the
reference pitch value (output of the controller) is increased, thus not allowing the
blades to reduce its pitch angles, which would further increase the RBMs. The change
of the reference pitch angle is proportional to the difference between the RBM
feedback and the constraint.

• Lines 468-470: Are the 'quasi-steady loads' determined by dynamic simulation
with uniform and constant wind speed? That is not the same as quasi-steady
(even though the outcome might be similar). Could the procedure for this
assessment be described with a little bit more detail?

This term describes simulations with steady and uniform inflow, steady state
operation (steady pitch and rotational speed), including elastic deformations on the
turbine structure. As also suggested by referee No. 2, we changed the wording to
"steady-uniform inflow loads".

• Lines 478-480: This statement seems to contradict the earlier description. Does
this only apply to the tip deflection? Why wouldn't the same argument apply to
flapwise RBM and thrust?

Indeed, we investigated two wind speeds ($v_{rated}$ and $v_{shift,start}$) for the steady-inflow cases and choose to display the more severe load case. For the flapwise RBM it doesn't matter since the load level is the same for the two wind speeds as defined by the design methodology. For the edgewise RBM, the thrust and the tower base bending moments it is the load case at $v_{rated}$. For the tip-to-tower-clearance it is $v_{shift,start}$. We clarified this in the revised manuscript:

First, the white bars illustrate the maximum loads under steady and uniform inflow including elastic deformations. Two wind speeds (rated and $v_{shift,start}$) were investigated and the more severe case is displayed here.

- Lines 487-488: This describe the normalisation of the loads. It doesn't mention that a different normalisation is used for operational load cases and storm load cases. It would be useful to mention this up front, to avoid confusion with interpretation of the results later. This use two different normalisation values might even be reconsidered, even though I can see arguments for its use. Nevertheless, in the discussion and conclusions the authors now need to warn the reader that values for operational load cases and storm load cases cannot be compared directly. On line 678, they state that this is due to using relative values, but it is actually due to using different reference values for each.

We see the disadvantages of using normalized load levels and choose to display only absolute values for both, the *Hybrid-Lambda Rotor*, as well as for the reference turbine. The updated Fig. 15 (appended to this authors' response) now includes more information about the distinctive load levels. We further decided to show the tip-to-tower-clearance instead of the out-of-plane deflections since this variable is more design driving. The descriptive pats in the manuscript (Sect. 3.4.3) are updated respectively.

Figure 15 presents the ultimate loads of the *Hybrid-Lambda Rotor* with solid bars and those from the reference turbine with hatched bars. Three groups are distinguished by their texture. First, the white bars illustrate the maximum loads under steady and uniform inflow including elastic deformations. Two wind speeds (rated and $v_{shift,start}$) were investigated and the more severe case is displayed here. Second, the grey bars show the theoretical load increase according to the generic scaling law as described by Gasch and Twele (2012), which would apply to a geometrically scaled reference turbine without changing the aerodynamic concept (e.g. scaling the steady-inflow loads of the IEA 15 MW, displayed with white hatched bars). (…) The unloaded tip-to-tower-clearance scales with $n$, too (neglecting gravitational effects). Thus, the loaded tip-to-tower clearance scales with $n$ as it is the difference of two variables, both scaling with $n$ (the unloaded tip-to-tower clearance and the maximum tip deflection with the blade in front of the tower). These scaling factors are only an indication for the upper bound since the design methodology of the *Hybrid-Lambda Rotor* includes peak shaving with a constant flapwise RBM. Third, the coloured columns relate to the dynamic load quantities from aero-servo-elastic simulations. (...)

The tip-to-tower-clearance represents a reserve, thus a higher value indicates a safer design. Note, that the unloaded tip-to-tower clearance also increased as documented in Table 2. The loaded tip-to-tower-clearance is larger for the *Hybrid-Lambda Rotor* in steady-uniform inflow as expected by the scaling law. (...)

The objective of the *Hybrid-Lambda Rotor* is to limit the stationary flapwise RBM to the maximum value of the reference turbine in steady-inflow BEM simulations. Thus, it is of special interest how much this type of loading increases in transient aeroelastic simulations. The ultimate load from normal power production is indeed marginally increased compared to the load level of the reference turbine from normal power production. But, if compared to the load level of the reference turbine under extreme wind shear events, the increase is only marginal.

- Lines 500-501, 504-505, (680-682,) 734 and 738: It is stated that the increase in DLC 6.3 is significant compared to the reference turbine. If I'm correct, this is confusing if not misleading, since DLC 6.3 is not assessed for the reference wind turbine. After this observation of increased loading, it is nevertheless claimed that the slender blade design shows benefits (= load reduction?) in storm events. This is also confusing. Perhaps it is meant that the increase in loading is not as large as it would have been in case no slender blade design was used. However, this is not what is compared here (a Hybrid-lambda rotor and a conventionally upscaled rotor). Along similar lines, on line 738, it is concluded that the Hybrid-lambda rotor shows advantages in reducing loads. Especially here, out of context, this seems somewhat misleading. In absolute sense, the loads are not reduced. I probably agree with the point that might have been intended, if it is about combating the load increase with the design. Could this be rephrased?

As we show absolute values in the revised manuscript the addressed paragraphs are rephrased. It now also gets clear, that the flapwise RBM increases for the storm events compared to the reference turbine, but the absolute values are still below the load level from DLC 1.5 and 1.6.

- Lines 522-525: I agree with the effect of the longer tower (higher lever arm, for almost equal thrust). However, the second argument seems flawed to me. Soft towers have a lower dynamic amplification factor for excitation frequencies that are above the natural frequency. They can have larger displacements with the same or even lower (internal) moments, which is why they are 'soft' (low stiffness). Thus, the effect of softness is more complicated and can go either way (depending on the excitation frequencies).

We thank the referee for the rectification and removed the respective sentences:

The tower base fore-aft bending moment is increased for the *Hybrid-Lambda Rotor* in
the dynamic load cases although it is constant for the steady-inflow cases which
highlights the importance of investigating transient effects.

• Lines 574-576: This statement seems in line with visual observations from the
graph. However, the lever arm is increased for the Hybrid-lambda rotor, while
it decreases for the reference turbine. Doesn't that correspond to an increased
contribution of the outer part?

We agree that ideally one would see a decrease in the non-dimensional lever arm also
for the *Hybrid-Lambda Rotor.* However, we would like to put emphasis on the reduced
loading and reduced load overshoot during the transient event. We changed the
wording accordingly:

For the *Hybrid-Lambda Rotor,* the characteristic kink in the force distribution leads to
lower maximum out-of-plane forces per unit length, even in the transient case. The
non-dimensional lever arm is only slightly increased during the event and still much
lower than for the reference turbine.

• Line 595: I suggest removing the reference to the aspect of market value here.
At this point (the model for) market value is not yet introduced to the reader.

As suggested, we removed the aspect of the market value from this paragraph since
it is further mentioned in line 673.

Figure 19 shows the gross energy yield per wind speed bin together with the Weibull
distribution of the cluster-wake affected reference site
.

• Line 611-612: To some extent the limitation of the flapwise RBM will oppose
this effect of geometric scaling. Although I agree that the mass will increase
stronger than for the Hybrid-lambda rotor, it doesn't seem fair to model the
structure of the Hybrid-lambda rotor and only hypothesise for more
conventional scaling. Furthermore, line 359 states that the mass of the new
blade is only 14% lighter than that of a scaled blade. Is 14% considered to be
'strongly increased'?

Since this paragraph is about AEP and revenue and not about blade mass and costs, we removed the argument using the blade mass.

Considering the cluster-wake affected wind speed distribution, the AEP can be increased by 3% and the economic revenue by 4%.

- Figure 18: Why are results shown for the non-optimised tower? The optimised design seems to be the only sensible design, which fulfils the constraints with the actual (quasi-steady) loads.

We apologize for the typo in the legend of Fig. 20. What is compared here is the IEA 15 MW, the initial blade and tower design and the optimized blade and tower design. Thus, this graph should highlight how much of the benefits result from the application of the "raw" *Hybrid-Lambda* design methodology and how much it can further be improved by the structural optimization. As suggested in a previous comment, we further added LCOE to the Figure.

***Technical corrections***

- Overall: 'Sec.', 'Sect.' and 'section' are used, without consistency. Same for 'Fig.' and 'Figure'.

We follow the author guidelines for WES journal papers (https://www.wind-energy-science.net/submission.html):

<Cite>

*"The abbreviation "Fig." should be used when it appears in running text and should be followed by a number unless it comes at the beginning of a sentence, e.g.: "The results are depicted in Fig. 5. Figure 9 reveals that…".*

*The abbreviation "Sect." should be used when it appears in running text and should be followed by a number unless it comes at the beginning of a sentence.*

<end cite>

We corrected the abbreviation Sec. to Sect.

- Line 135: Considering line 175-176, probably 'adjusted' is meant here. 'Adopt' implies that it is kept the same (in dimensionless spanwise coordinates). Alternatively, it could have been meant that the same 'order' was adopted, instead of the distribution.

The airfoil distribution is adopted (kept the same in dimensionless spanwise coordinates) in a first step and then optimized in the structural optimization process.

As the *Hybrid-Lambda Rotor* is compared with the IEA 15 MW reference turbine, the same airfoil family is used and the airfoil distribution along blade span is adopted in a first step. The airfoil position is later optimized as described in Sect. 2.3.

- Line 170: The use of 'maximum' is confusing here (especially for a low-induction rotor, which doesn't operate at maximum power coefficient in design conditions). Is it meant at TSR 11 (and at which pitch)?

We are referring to the maximum power coefficient for the given turbine design over all TSR and pitch (in this case at TSR=11 and at fine pitch=-0.8°).

The wind speed at which the transition from the light wind to the strong wind mode should start is calculated first. This is done by finding the operational point at maximum power coefficient for the given turbine design (at $TSR = 11$ and $fine\ pitch = -0.8°$) when the limiting flapwise RBM is first reached.

- Line 187: 'choice for' would be more appropriate than 'assumption of'. The authors are not addressing an unknown aspect here.

As the main focus of this paper is the aerodynamic rotor concept, the simple  choice of a monopile foundation was made ...

- Line 201: 'planed' -> 'planned'.

... further simulations are  planned using ...

- Line 213 and 215: 'blade design' -> 'aerodynamic blade design'.
- Line 220-221: 'which ... moments' would be more appropriate as an argument on line 151.
- Line 273: Probably 'that' is meant, instead of 'which'.

Incorporated the demanded changes from the three mentioned bullet points.

- Line 228 (Heading section 3.3): It is not the 'model' that is designed and optimised.

Changed the heading to:

Optimization of the structural blade and tower design

- Line 357: 'up' -> 'down'.

Thank you for pointing out this typo. We incorporated this important detail.

- Line 530: 'The unsteady event [add: starts after 200 seconds and] lasts for 12 seconds, ...'.

The unsteady event starts at 200 seconds and lasts for 12 seconds with a maximum wind speed at the top of the rotor disc after 6 seconds.

**Referee 2:**

The manuscript presents a design and optimization methodology for a novel wind turbine rotor concept the authors call 'Hybrid-Lambda'. The work aims to a design rotor where (i) the outer part of the rotor is set to be optimal at low wind speeds operating at high TSR, and the inner part is designed for higher wind speeds at a lower TSR, and (ii) the increased loads are managed through a peak shaving controller close to rated conditions. The authors target to achieve this while constraining the mean blade flapwise bending moment loads below the max value of the reference turbine (IEA 15MW). As stated by the authors, the economic motivation for the design is to take advantage of energy pricing at low-wind conditions.

The work presented is scientifically significant and proves to challenge the conventional design of horizontal axis wind turbine rotors. The motivation and objectives of the work is presented clearly. But when presenting the methodology and results the ideas/concepts/fundamentals are difficult to follow. I do acknowledge that the body of work presented here is immense and there are a lot of moving parts to the novel rotor design. Light restructuring of concepts will help the readers appreciate the value of the manuscript. As an example, moving the controller strategy outlined in section 3.2 and figure 6 to line-125 would strengthen Section 2.

We thank the referee for the constructive and positive feedback.

We moved Fig. 2 to Sect. 2 and added two descriptive paragraphs to Sect. 2.1 to explain the rotational speed schedule and the peak shaving up front.

The transition between the operating modes introduces a new control region since the switching of the TSR is not a sudden change rather than a continuous reduction in TSR. In this paper, it is realized with a constant rotational speed (rpm) in region 2.2 as shown in Fig. 2. The reduction in TSR alone (with a constant rpm) is not enough to limit the loads. On the contrary, it is part of the design methodology to combine pitching to feather and a reduction in TSR for load limitation as further analysed in Sec. 3.1. Consequently, the so-called strong wind mode cannot be described with a constant pitch angle. With increasing wind speed the pitch angle is gradually increased towards feather to limit the flapwise RBM. This action will be referred to as peak-shaving in the following.

Note, that the transition of TSRs could also be realized in different ways (e.g. reduction or gentle increase in rpm). In fact, the optimal combination of TSR and pitch for the transition region can be found by constraining the flapwise RBM and searching for the optimum in the power coefficient. These optimization routines resulted in a gently increasing rpm until rated wind speed. However, for all wind speed bins, the increase in the power output was never larger than a tenth of a Megawatt compared to the constant rpm solution presented here. Consequently, the aforementioned alternative for the transition region is not presented in this paper.

Overall, the manuscript is well structured and provides significant work that will be
valuable to the broader wind energy community. Detailed comments and minor
corrections are shared below:

Detailed Comments:

1.  Section 1, line 58-59: Similarity to Wobben's work is presented, but it is not clear
how the current work differentiates from itself until section 4. Please include
details on how this work sets itself apart from previous works in the
introduction.

We moved the description to the introduction, as proposed.

This concept follows the objective of reducing unintended stall effects on the blade of
a variable-speed turbine in gusty winds. It was not used to enable large rotors with
low specific ratings, as pointed out with the *Hybrid-Lambda* concept.

2.  Section 1, line 63-64: I do not agree with the terminology "zero pitch" used in-
lieu of "fine pitch". Typically, the blade tip is set to a pitch angle of zero and is a
reference orientation for the geometric twist of the blade. "Fine Pitch" is the
additional pitch offset added during operation such that the tip of the blade is
at the optimal design twist. Please make the necessary changes here and
through the manuscript.

We added the definition of the term fine pitch and made the necessary changes
throughout the manuscript. The term zero pitch should indicate only that the pitch
angle is at zero degrees. In many blade design studies the so called fine pitch, that
leads to the maximum power coefficient at design TSR, can deviate slightly from zero
degrees, as it does in our study. We clarified this and added to the description of Fig. 2:

From 4 $m\,s^{-1}$ on, the rotor operates at the high TSR of 11 in the light wind mode and
a fine pitch angle of -0.8° which leads to the maximum power coefficient. This pitch
angle is called fine pitch since the pitch angle for optimal $c_p$ was derived after the
blade design was concluded.

3.  Section 2.1, line 119-120: The concept of peak-shaving is introduced but it is
not clear what the procedure entails. Please provide a brief description.

We added a definition of the term peak shaving as printed in Sect. 2.1.

The transition between the operating modes introduces a new control region since
the switching of the TSR is not a sudden change rather than a continuous reduction
in TSR. In this paper, it is realized with a constant rotational speed (rpm) in region 2.2
as shown in Fig. 2. The reduction in TSR alone (with a constant rpm) is not enough to
limit the loads. On the contrary, it is part of the design methodology to combine
pitching to feather and a reduction in TSR for load limitation as further analysed in
Sec. 3.1. Consequently, the so-called strong wind mode cannot be described with a
constant pitch angle. With increasing wind speed the pitch angle is gradually increased towards feather to limit the flapwise RBM. This action will be referred to as peak-
shaving in the following.

4. Please comment on how peak shaving influences the design of the blade.
Reading though the manuscript, it feels like a control strategy and not
something influencing the aerodynamic design of the rotor.

We carried out an additional study and applied the *Hybrid-Lambda* control strategy to
a conventionally scaled blade. This breaks up to what extend the benefits result from
the adjusted control strategies and to what extend they result from the adjusted blade
design. In fact, they go hand in hand. A major part results from the control
optimization and the TSR will be reduced over a wide range of wind speeds. Thus, it
makes sense to account for that fact in the blade design. We added the results to Fig. 9
and added a descriptive paragraph:

The green dashed line indicates the power curve of the reference blade that is
geometrically scaled by the same factor and conventional peak shaving is applied to
limit the flapwise RBM. This means only the pitch angle is set to a higher value to
constrain the flapwise RBM while the rpm follows the design TSR. In contrast, the black
dotted line represents the same blade (geometrically scaled IEA 15 MW) but peak
shaving is applied in a similar manner as for the Hybrid-Lambda Rotor. This means
for $v > v_{shift,start}$ the rpm is kept constant until the operational TSR is reduced from 9
to 7. For $v > v_{shift,end}$ the rpm schedule follows the TSR of 7 which is an arbitrary
choice in this case and should be optimized in a detailed design study. In addition, the
pitch angle is set for $v > v_{shift,start}$ in order to limit the flapwise RBM. In short, we are
applying the Hybrid-Lambda control strategy to a conventional blade design. The
results show that the power output can be greatly increased if the TSR is lowered in
region 2.2 and 2.3 (compare green dashed and black dotted line in Fig. 9). Thus, peak
shaving should not only be accomplished by increasing the pitch angle, but also by
optimizing the operational TSR with respect to the load constraint (as also indicated
by Madsen et al. (2020)). Since the results show that a reduction of the operational
TSR is beneficial in the peak shaving region it makes sense to account for this fact
already in the blade design which is integrated in the Hybrid-Lambda design
methodology. Indeed, the Hybrid-Lambda Rotor enables even lower power losses in
the peak shaving region since the TSR reduction is already accounted for in the blade
design (compare solid red and dotted black line in Fig. 9). The turbine concept reaches
its rated power at 10.2 $m\ s^{-1}$, which is 0.4 $m\ s^{-1}$ lower than the reference turbine.

5. Section 2.3: In this section the free variables are defined as chord, twist, radial
airfoil positions, and spar cap thickness. But in section 2.1, the transition
position and rotor radius are also discussed as design variables. Please clarify
in the manuscript which variables are set/pre-determined and which ones are
free variables.

We clarified this by adding the type of variable (optimized, fixed…) to Table 2 and
additionally provide a design flow chart in the revised manuscript which clarifies the
design workflow.

The overall design and optimization workflow is illustrated in Fig. 3. The process can
be explained in four steps: An aerodynamic blade optimization, an aero-structural
optimization of the blade, a structural optimization of the tower and the aero-servo-
elastic simulations. In the first step (aerodynamic optimization), the design variables
are the transition point between the inner and outer blade section, the design TSRs,
the design axial induction factors, the twist offset and the design angle of attacks.
Once a reasonable design is established the influence of the rotor radius is
investigated. In the second step (the aero-structural optimization), the design
variables are the airfoil positions and the spar cap thickness. When this step is
converged the aerodynamic optimization is re-calculated once with the new airfoil
positions. As a third step, the tower and monopile are optimized for a fixed rotor
design. The resulting turbine design is then investigated in aero-servo-elastic
simulations.

6. Section 2.3: The load case for the optimization is defined at a wind speed of
6.9m/s, the following sentence on line 180-181 does not justify why this case
was selected. If the rotor radius is a free parameter, then, the inflow for the
load case is going to be a function of radius as the TSR is set to 11, this is
confusing. How was this predetermined?

The operational parameters for the design load case are re-calculated for every design
iteration in WISDEM (with the code changes applied by the authors). We added this
description to the methodology:

For each iteration the schedule of rpm, pitch, power, thrust and flapwise RBM over
wind speed is re-calculated. The considered load case for the constraints is a steady
inflow at the strongest wind speed in the light wind mode $v_{shift,start}$, as calculated for
each design iteration (in this case $v = 6.9\frac{m}{s}$, $TSR = 11$, $\beta = -0.8°$).

7. Section 2.3, line 204: OpenFAST provides a large set of options in its
aerodynamic module AeroDyn. Please elaborate on what aerodynamic options
were used when carrying out the aero-elastic simulations. Was it the same as
the reference wind turbine? This will help guide discussing the load
comparisons.

We added this information to Sect. 2.3 and further plan to provide the simulation
model of the *Hybrid-Lambda Rotor* once the manuscript is published. The aerodynamic
model was chosen the same way for the reference turbine.

The aerodynamic modelling includes the effects of tower shadow and the
aerodynamic loading on the tower, as well as the Minemma/Pierce dynamic stall
model, as described by Damiani et al. (2019).

8. Section 3.1, line 216-217: It is not clear how the specific rating and rotor diameter is determined? Was it a design variable? If so, please define in Section 2.2/2.3. If not, please clarify on how this was determined.

The influence of the rotor diameter is investigated in a subsequent design loop once an initial chord and twist distribution is established. This is also marked in the newly added design flow chart. The influence of the rotor diameter as a design variable is discussed in line 279 of the revised manuscript. To simplify the understanding, the concept can only be shown for one specific rotor diameter in the given paper. To clarify this, we classified the rotor diameter as a design variable in Table 2 and added an explanation to the beginning of Sect. 3.

In this chapter, we focus on the given use-case of the 15 MW offshore wind turbine, no longer generalizing the concept, in order to simplify the understanding. This means only one specific turbine diameter is presented here, although the influence of the rotor radius as a design variable was investigated and is further described below. We first address the resulting blade design and the influence of certain design variables. Table 2 summarizes general turbine parameters. The second part deals with loads, axial induction, angle of attack and power generation under steady and uniform inflow conditions. This is followed by the results of the structural design and the aero-servo-elastic investigations.

9. Section 3.1, line 221-222: I find it difficult to follow the need for the twist offset in the inner section of the blade. The discussion related to this in previous and future sections feel fragmented. Please try re-organizing and better explain the need for the twist offset.

Indeed, the arguments and explanations were fragmented over several sections. We re-organized the description of the aerodynamic behaviour and the change in the angle of attack distribution to bundle the arguments. We moved the description of the change of the inflow angle distribution due to the change in TSR (lines 338-353 and Fig. 7), to Sect. 3.2 (next to Fig. 8). Like this, the description of inflow angle change and angle of attack distribution follow up on each other and are easier to understand.

Regarding the twist offset, we added a description to Sect. 2.1.

Furthermore, we design the blade in a way that peak shaving is applied more efficiently. The inner section is designed with a twist offset towards stall. This comes with several advantages. The inner section does not operate in the design point in the low wind regime. As it is twisted towards stall and operated at a higher TSR than it was designed for, a fairly conventional induction factor of 0.33 can be reached, which leads to an increase in the power coefficient in the low wind regime. The angle for the twist offset is derived iteratively in stationary blade element momentum (BEM) simulations to reach the desired axial induction factor of 0.33 in the inner section at the high TSR. Using the twist instead of the chord length as a tool for this increase in the axial induction factor allows to use smaller chord lengths which leads to more slender, lighter and possibly cheaper blades. Hence, the twist offset defines the difference of the axial induction factor between the light and strong wind mode for the inner part of the blade and it further influences the pitch angle at $v_{shift,end}$ that is needed to limit the loads. In fact, the pitch angle of 2.2° at $v_{shift,end}$ almost perfectly counterbalances the twist offset of -2.5°. Hence, the inner part of the blade operates in it's optimal lift to drag ratio at this wind speed, although the entire blade is already pitched to feather for load reduction. When peak shaving is applied, pitching shifts the inner section to operate at its aerodynamic optimum rather than moving away from it. It reaches its design point (an induction factor of 0.21 at the low TSR), which is beneficial for load reduction. In contrast, the outer section is now operated in a "pitched-to-feather-condition" and is greatly relieved. The limits to this methodology are negative lift and the stall angle. The latter is also plotted in Fig. 8.

10. Section 3.1, line 254-269: This paragraph emphasis and extensively discusses the rotor radius as a varying parameter, this leads the reader to believe that it is a design parameter, but it has not been highlighted as such in Section 2.3.

We classified the rotor diameter as a design variable in Table 2 and added an explanation to the beginning of Sect. 3.

In this chapter, we focus on the given use-case of the 15 MW offshore wind turbine, no longer generalizing the concept, in order to simplify the understanding. This means only one specific turbine diameter is presented here, although the influence of the rotor radius as a design variable was investigated and is further described below. We first address the resulting blade design and the influence of certain design variables. Table 2 summarizes general turbine parameters. The second part deals with loads, axial induction, angle of attack and power generation under steady and uniform inflow conditions. This is followed by the results of the structural design and the aero-servo-elastic investigations.

11. Section 3.1, line 276: This is the first time the transition point for lambda is presented as a design choice and not a free variable. There are a lot of variables and moving parts in the optimization to follow. Presenting the optimization/design workflow in a flow diagram would help guide the reader through the whole optimization process better, in fact it will help the authors be more clear in their discussion of the optimization process. Using XDSM (eXtended Design Structure Matrix) might be a good approach.

We thank the referee for the idea of visualizing the workflow in a design flow chart. We included this in Sect. 2.1 and added a descriptive paragraph.

The overall design and optimization workflow is illustrated in Fig. 3. The process can be explained in four steps: An aerodynamic blade optimization, an aero-structural optimization of the blade, a structural optimization of the tower and the aero-servo-elastic simulations. In the first step (aerodynamic optimization) the design variables are the transition point between the inner and outer blade section, the design TSRs, the design axial induction factors, the twist offset and the design angle of attacks. Once a reasonable design is established the influence of the rotor radius is investigated. In the second step (the aero-structural optimization) the design variables are the airfoil positions and the spar cap thickness. When this step is converged the aerodynamic optimization is re-calculated once with the new airfoil positions. As a third step the tower and monopile are optimized for a fixed rotor design. The resulting turbine design is then investigated in aero-servo-elastic simulations.

12. Section 3.2, line 289: Presenting figure 6 in section 2.1, around line 125 would help the readers better understand the unique speed and pitch schedule, and peak shaving that is discussed extensively up until line 289.

We moved Fig. 2 to Sect. 2.1, as suggested and added a description of the TSR-transition.

The transition between the operating modes introduces a new control region since the switching of the TSR is not a sudden change rather than a continuous reduction in TSR. In this paper, it is realized with a constant rotational speed (rpm) in region 2.2 as shown in Fig. 2. The reduction in TSR alone (with a constant rpm) is not enough to limit the loads. On the contrary, it is part of the design methodology to combine pitching to feather and a reduction in TSR for load limitation as further analysed in Sec. 3.1. Consequently, the so-called strong wind mode cannot be described with a constant pitch angle. With increasing wind speed the pitch angle is gradually increased towards feather to limit the flapwise RBM. This action will be referred to as peak-shaving in the following.

13. Section 3.2: Please discuss the limitations of using BEM specifically for the hybrid-lamda rotor. Given the step change in induction at the 70% blade span. Does using higher fidelity method like free-vortex or CFD change the load distribution near the 70% blade span?

We used free-vortex wake methods to investigate to what extend the assumption of independent blade elements in the BEM theory is violated. This is addressed in a separate publication which is accepted but not published yet (Ribnitzky, Bortolotti, Branlard, Kühn: *Rotor and wake aerodynamic analysis of the Hybrid-Lambda concept - an offshore low-specific-rating rotor concept,* JoP conference series, 2023). The FVW investigations support the design principles of the *Hybrid-Lambda Rotor* that were originally identified using the BEM theory. The integrated rotor quantities (power and thrust) are in very good agreement for the two methods. For the light-wind mode, the aerodynamic power exactly matched, whereas the FVW code computed about 0.5% higher thrust. For the strong-wind mode, the FVW code computed 1.5% higher power and 0.75% higher thrust. For the radially resolved variables, discrepancies are most distinct when the gradients along the blade span are large. In the light-wind mode, differences of about 0.03 in the axial induction factor distribution are observed between BEM and FVW. In the strong-wind mode, the deviations are less prominent, as the gradients along the blade span are reduced. We included the citation in the beginning of Sect. 3.2:

Note, that due to the gradients along the blade span the assumptions made in the
BEM theory can reach their limit. We used free-vortex wake methods to investigate to
what extend the assumption of independent blade elements in the BEM theory is
violated. Results show good agreements for rotor integrated quantities (power and
thrust), although some differences are noticeable in the radius resolved variables
when the gradients along the blade span are large in the light wind mode. The
interested reader is referred to Ribnitzky et al. (2023).

14. Section 3.3, Line 350: In addition to presenting the relative thickness and the
spar-cap thickness, it would be valuable to compare the flapwise and edgewise
stiffness, and mass distribution of the blade vs the IEA 15MW. The rapid
transition in stiffness at the 70% location of the blade will be a point of concern
especially for extreme loads. The optimization routine uses a steady inflow
condition at relatively low wind speeds (as discussed in Section 2.3) this will not
be representative of the stiffness distribution at the TSR transition region of the
blade.

We added a plot, comparing the mass and stiffness distribution of the *Hybrid-Lambda*
blade and the IEA 15 MW:

The resulting mass and stiffness distributions are compared to those of the IEA 15 MW
in Fig. 13, clearly showing the steeper gradient in the flapwise stiffness in the transition
area of *the Hybrid-Lambda* blade.

15. Section 3.3, Line 361-362: Using an exponent of 3 for geometrically scaling the
reference blade for comparison is unfair. More recent publications (Griffith
2014, SNL100-03) have shown that the mass scaling exponent is realistically
between 2.1 to 2.5.

We added a note with the suggested citation:

Note, that the reference exponent of 3 is only derived by geometric considerations.
Griffith and Richards (2014) summarize recent trends for commercial and research
blades and state mass scaling exponents of 2.5 for moderately innovative blades and
2.1 for highly innovative designs.

16. Section 3.3, Line 366: What is the tower design driver for the IEA 15MW turbine?
How does that contrast to the design driver for the current design? The
reduction in tower diameter from 10m to 8.54m is significant especially given
the 13% increase in blade mass (based on Line 362).

The optimization algorithm reduced the tower diameter but increased the wall
thickness (as described in line 442) in order to meet the constraints for buckling,
maximum stress and eigenfrequencies. Furthermore, we increased the partial safety
factor for loads (see line 230), to account for the simplified load analysis. We further
want to point out that the thrust of the *Hybrid-Lambda* turbine is lower than for the
IEA 15 MW in all DLCs (as can be seen in Fig. 15). Nevertheless, the increased rotornacelle-assembly mass, the resulting dynamic loads and the storm loads will lead to a challenging load set, that needs to be taken care of when deriving a sophisticated tower design. We would like to focus on the rotor design in this paper and chose to present a very simplified tower design. We added to line 223:

As the main focus of this paper is the aerodynamic rotor concept, we only present a preliminary tower design and the simple choice of a monopile foundation was made, although...

17. Section 3.4.1, Line 390: The equation is typically used for a constant Cp region. Since this value is not unique for the hybrid-lambda rotor how is the generator torque determined?

We implemented the desired $c_p$ value as a function of rotational speed which is derived from steady state simulations. We added the dependency in the equation and added:

$$M_g = \frac{\pi R^5 \rho c_p(\omega)}{2\lambda^3} \omega^2$$

Note, that there is no unique $c_p$ in region 2.3 since the pitch angle is a function of wind speed. Hence, the desired $c_p$ from steady state simulations is implemented as a function of rotational speed.

18. Section 3.4.2: Can you comment on the increased pitch activity due to the newer controller as compared to the reference? This will be important when determining the scaling of components (like pitch bearing/pitch actuator) costs for the final cost function.

In the initial manuscript, we missed to comment on the pitch activity and included a statement in line 494:

In this way, the amplitude of load variations can be drastically reduced and load overshoots are less severe. Nevertheless, the increased pitch activity needs to be considered when sizing the actuators and bearings which will influence the resulting cost function.

19. Section 3.4.3: Does 'quasi-steady loads' refer to the loads experienced by the turbine due to steady inflow? If so please replace with 'steady state loads' or 'steady-inflow loads'.

Yes, with quasi steady loads we want to describe the loads from simulations with steady and uniform inflow, including elastic deformations of the structure. We changed the wording to "steady-inflow loads" throughout the manuscript. If only rigid structures are considered, this is additionally mentioned.

20. Section 3.4.3, Line 504-505: In storm cases, it is not only the slenderness of the blade that determines the load or reduction in loads. It is the complex

| 1167 | interactions arising due to the blade geometrical twist, azimuthal angle, and |
| 1168 | yaw error that determines the loading of the turbine. Attributing the lower |
| 1169 | storm loads to planform area is assuming the inflow to the blades are primarily |
| 1170 | in at 90-deg to the airfoils, this is far from the case. |

We do agree that this formulation was misleading, so we reformulated it.

In the storm events, the slender blade design shows additional benefits. The shorter
chord length reduces the lift forces arising from the complex interaction of blade twist,
azimuthal position and yaw error.

21. Section 3.5: Generally, any discussions regarding CapEx increases/decreases in
components other than blade/rotor and tower are neglected. It will add value
if the authors share why CapEx change of other components are significant (or
not) to COVE.

We caried out an additional study on the component costs using the cost models
implemented in *WISDEM*. We added the description of the methodology to Sect. 2.3:

The cost model implemented in WISDEM based on the work from Fingersh et al. (2006)
was used to create a breakdown of the costs of major wind turbine components. The
model includes a rather detailed estimation of the blade costs, as described by
Bortolotti et al. (2019), including assumptions for materials, labour, tooling and many
more aspects. On the contrary, the costs for parts like the pitch system and the hub
are implemented as simple functions of the rotor diameter or the blade mass. The
assumption of the direct drive generator costs was adjusted since the original model
only takes the machine rating as an input. In our case, the rated power remains
constant but the rated torque increases since the maximum rpm is reduced (constant
maximum blade tip speed). According to Fingersh et al. (2006), the generator mass
scales with $M_{g,rated}{}^{0.606}$, with $M_{g,rated}$ being the rated generator torque. We accounted
for the mass increase in the cost estimation, assuming that the costs increase linear
with the mass. Overall, the cost model can serve to point out trends in the
development of costs when increasing the turbine size, but absolute values should be
handled with care.

We added a new bar chart with the cost breakdown to Sect. 3.5 and described the
results:

A breakdown of the costs for the most important turbine components is shown in
Fig. 21. Obviously, the largest increase in costs compared to the reference turbine is
seen for the blades, since this is the part that increased the most in terms of size and
complexity. In fact, the costs of a blade increased by a factor of 2.8 (equals $n^{3.37}$).
Related to the much heavier blades and the increased aerodynamic loading also the
pitch system needs to be sized properly. Hence, the pitch system (plotted for all three
blades) sees the second highest increase with a factor of 1.8, compared to the
reference turbine. The tower costs increased by a factor of 1.2. The costs for the direct
drive generator have the largest share of the total turbine costs and the derived generator costs for the reference turbine are comparable with the findings of Barter
et al. (2023). For the *Hybrid-Lambda Rotor*, they increased by a factor of 1.22 since the
rated generator torque increased. These numbers should only indicate an
approximate trend of the cost breakdown, since the cost model in *WISDEM* relies on
simplified scaling rules coupled to empirical datasets. For more insights, sophisticated
models need to be set up for components like the pitch and yaw system or the
generator.

22. Section 4: This section generally reads well.

23. Section 4, 656: The authors contrast their work to that of Wobben, please
consider moving this discussion to the literature review to make a stronger
argument about the novelty of the Hybrid-Lambda rotor.

We moved the description to the introduction, as proposed.

This concept follows the objective of reducing unintended stall effects on the blade of
a variable-speed turbine in gusty winds. It was not used to enable large rotors with
low specific ratings, as pointed out with the *Hybrid-Lambda* concept.

24. Section 4, 665-666: what does "way more than 100m length" mean in this
context? Is it a mis-phrased sentence?

We re-phrased the sentence:

Thus, we want to raise the question of whether controlling one degree in the angle of
attack is at all feasible in a real application of a blade with 158 m length.

25. Section 4, 685-670: Yes, I strongly agree with the authors the value of
considering the torsional degree of freedom for the blade. Especially given its
slender nature. Consequently, the aero-elastic stability of the blade will be
interesting given how close to stall the inner part of the blade is at certain
operational conditions.

We do agree with the referee. Unfortunately, we feel that considering blade torsion in
the analysis will only make sense in combination of a major redesign of the blade since
the torsional deflection needs to be accounted for in the blade design. Further, a full
aero-elastic stability analysis would go beyond the scope of this paper which aims on
providing the conceptual idea and the methodology to design very low-specific rating
wind turbines.

26. Section 5, lines 710-712: After reading the paper it is not yet clear to me how
the peak-shaving is integrated into the design process of the rotor, or how the
aerodynamic parameters are influenced by it. The aforementioned flow
diagram for the design/optimization process will help guide the reader to this
conclusion.

As suggested, we included the design flow chart in the revised manuscript. Further, the additional study on applying the *Hybrid-Lambda* control strategies to a conventionally scaled blade will provide more evidence that the peak shaving control strategies and the changes in the blade design go hand in hand.

Minor corrections:

1. Line 157: Citation for Buhl might be missing.

We added the respective citation:

Buhl, L.: A New Empirical Relationship between Thrust Coefficient and Induction Factor for the Turbulent Windmill State, National Renewable Energy Laboratory, NREL/TP-500-36834, 2005.

2. Line 170: The source code ..... as described in the following (sections).

We changed the wording:

... as described here.

3. Line 201: Typo, 'planed'

We corrected the typo.

4. Section 3 title: 'Design and optimization of the blade structure'?

Section 3 covers all the results and the following subsections:

- Aerodynamic blade design
- Aerodynamics, loads and power under steady-inflow BEM simulations
- Optimization of the structural blade and tower design
- Aeroelastic load simulations
- Techno-economic evaluation

We therefore would like to keep the very generalized heading of "Results" for Sect. 3.

5. Line 451: avoid using the word 'slight' when discussing quantitative values like RBM.

We replaced the word "slight" with "minor" or "marginally" throughout the manuscript.

6. Line 593: 'Figure' is used to reference figure 17, whereas in the previous sections 'Fig. XX' has been used. Please maintain consistency.

We follow the author guidelines for WES journal papers as further described in the
respond to the first reviewer (first comment in "technical corrections"). Figure and
Section are not abbreviated if it comes at the beginning of a sentence.

7. Line 599: 'Sect. 1' is used to refer to a Section, whereas 'Sec. XX' was used
   previously. It is clear that different authors have contributed to the sections,
   hence the change in style, but please maintain constancy throughout the
   manuscript as it is a single body of work.

We replaced the abbreviation "Sec." with "Sect." to follow the author guidelines for
WES journal papers.

**Appendix:**

[Figure]

**Figure 3:** Design and optimization work flow of the *Hybrid-Lambda* concept, round bullet points: Free design variables,
squared bullet points: Constraints, diamonds: Outputs, f(...) : As a function of (...), LW: Light wind, SW: Strong wind

[Figure]

**Figure 9:** Power output of the *Hybrid-Lambda Rotor* (solid red) compared to the reference turbine (blue) and a scaled
reference turbine (dashed green and dotted black)

[Figure]

**Figure 13:** Mass and stiffness distribution for the *optimized Hybrid-Lambda blade* (red) and the IEA 15 MW (blue)

[Figure]

**Figure 15:** Ultimate loads in solid bars for the *Hybrid-Lambda Rotor* and in hatched bars for the IEA 15 MW reference
turbine, only critical loads are displayed, EWSH = extreme wind shear horizontal, EWSV = extreme wind shear vertical, f-a
BM = fore-aft bending moment, s-s BM = side-side bending moment

[Figure]

**Figure 20:** Reduction in cost of valued energy and LCoE relative to the reference turbine for the cluster-wake affected wind
speed distribution

[Figure]

**Figure 21:** Estimation of turbine component costs for the IEA 15 MW (blue) and the optimized *Hybrid-Lambda* turbine (red)

**References:**

Burton, T., Jenkins, N., Sharpe, D., and Bossanyi, E.: Wind Energy Handbook, Wiley, Chichester, 2nd edn., 2011.

*Further references mentioned in the blue citations can be found in the revised manuscript.*